# Mitochondrial calcium uptake orchestrates vertebrate pigmentation via transcriptional regulation of keratin filaments

Jyoti Tanwar[1], Kriti Ahuja[1], Akshay Sharma[1], Paras Sehgal[2], Gyan Ranjan[1], Farina Sultan[1¤], Anushka Agrawal[1], Donato D'Angelo[3], Anshu Priya[2], Vamsi K. Yenamandra[2], Archana Singh[2], Anna Raffaello[3], Muniswamy Madesh[4], Rosario Rizzuto[3,5], Sridhar Sivasubbu[2], Rajender K. Motiani[1] *

1 Laboratory of Calciomics and Systemic Pathophysiology (LCSP), Regional Centre for Biotechnology (RCB), Faridabad, Delhi-NCR, India, 2 CSIR-Institute of Genomics and Integrative Biology (IGIB), New Delhi, India; Academy of Scientific and Innovative Research (AcSIR), Ghaziabad, India, 3 Department of Biomedical Sciences, University of Padua, Padua, Italy, 4 Department of Medicine, Center for Mitochondrial Medicine, Cardiology Division, University of Texas Health San Antonio, San Antonio, Texas, United States of America, 5 National Center on Gene Therapy and RNA-Based Drugs, Padua, Italy

¤ Current address: CSIR-Indian Institute of Integrative Medicine, Jammu, India
* rajender.motiani@rcb.res.in

**Data Availability Statement:** All relevant data are within the paper and its Supporting Information files. The raw RNA sequencing data have been

## Abstract

Mitochondria regulate several physiological functions through mitochondrial $Ca^{2+}$ dynamics. However, role of mitochondrial $Ca^{2+}$ signaling in melanosome biology remains unknown. Here, we show that pigmentation requires mitochondrial $Ca^{2+}$ uptake. In vitro gain and loss of function studies demonstrate that mitochondrial $Ca^{2+}$ uniporter (MCU) is crucial for melanogenesis while MCU rheostat, MCUb negatively control melanogenesis. Zebrafish, $MCU^{+/-}$ and $MCUb^{-/-}$ mice models show that MCU complex drives pigmentation in vivo. Mechanistically, MCU silencing activates transcription factor NFAT2 to induce expression of keratin (5, 7, and 8) filaments. Interestingly, keratin5 in turn augments mitochondrial $Ca^{2+}$ uptake and potentiates melanogenesis by regulating melanosome biogenesis and maturation. Hence this signaling module acts as a negative feedback loop that fine-tunes both mitochondrial $Ca^{2+}$ signaling and pigmentation. Notably, mitoxantrone, an FDA approved drug that inhibits MCU, reduces pigmentation thereby highlighting therapeutic potential of targeting mitochondrial $Ca^{2+}$ uptake for clinical management of pigmentary disorders. Taken together, we reveal an MCU-NFAT2-Keratin5 driven signaling axis that acts as a critical determinant of mitochondrial $Ca^{2+}$ uptake and pigmentation. Given the vital role of mitochondrial $Ca^{2+}$ signaling and keratin filaments in cellular physiology, this feedback loop could be operational in a variety of other patho-physiological processes.

## Introduction

Mitochondria are robust signaling organelle that play a crucial role in physiology and disease [1–3]. One of the key signaling pathways that regulate mitochondria mediated control of cellular physiology is mitochondrial $Ca^{2+}$ dynamics [1,4,5]. The mitochondrial $Ca^{2+}$ signaling

deposited in the NCBI Sequence Read Archive (SRA) under the accession number PRJNA1112319.

**Funding:** This work was supported by the DBT/Wellcome Trust India Alliance Fellowship (IA/I/19/2/504651) awarded to R.K.M. R.K.M. acknowledges RCB Core Institutional Funding. The funders did not play any role in the study design, data collection and analysis, decision to publish, or preparation of the manuscript.

**Competing interests:** The authors have declared that no competing interests exist.

**Abbreviations:** DP, darkly pigmented; DPBS, Dulbecco's phosphate buffer saline; EMRE, essential MCU regulator; EPD, Eukaryotic Promoter Database; EV, empty vector; FBS, fetal bovine serum; GO, Gene Ontology; hpf, hours post fertilization; IVT, in vitro transcription; LD, low density; LP, lightly pigmented; MCU, mitochondrial calcium uniporter; NFDM, non-fat dry milk; OCR, oxygen consumption rate; SOCE, Store-Operated Calcium Entry; TCA, trichloroacetic acid; TEM, transmission electron microscope.

modulates several cellular functions including bioenergetics, autophagy, cytosolic $Ca^{2+}$ buffering, secretory functions, cell proliferation/migration, and cell survival [6,7]. Further, perturbations in mitochondrial $Ca^{2+}$ homeostasis lead to a wide range of pathological conditions such as variety of cancers, metabolic disorders, aging, and neurological conditions [6,8–10]. Therefore, it is critical for cells to maintain mitochondrial $Ca^{2+}$ homeostasis.

$Ca^{2+}$ uptake into the mitochondrial matrix is mediated via a highly $Ca^{2+}$ selective channel, i.e., mitochondrial calcium uniporter (MCU) [11,12]. The uniporter complex consists of pore-forming subunit MCU along with regulatory proteins namely mitochondrial $Ca^{2+}$ uptake proteins (MICU1/2), MCU regulatory subunit b (MCUb), and essential MCU regulator (EMRE) [4,13,14]. Wherein, MICU1/2 act as gatekeepers of MCU complex [4,13] and EMRE tethers MICU1/MICU2 to the MCU complex [15]. While MCUb acts as a dominant negative form of channel pore forming unit (MCU) and thereby it negatively regulates mitochondrial matrix $Ca^{2+}$ uptake [4,13]. On the other hand, mitochondrial $Ca^{2+}$ efflux is facilitated through Leucine zipper EF hand-containing transmembrane 1 (LETM1) [16] and $Na^+/Ca^{2+}$ exchanger-like protein termed as $Na^+/Ca^{2+}/Li^+$ exchanger (NCLX) [17]. Although key players involved in mitochondrial $Ca^{2+}$ influx and efflux are characterized, the signaling cascades that maintain mitochondrial $Ca^{2+}$ homeostasis remain poorly understood. Typically, nature embraces compensatory mechanisms or feedback loops for sustaining such crucial organelle function. Indeed, global MCU knockout mice have been reported to acquire compensatory mechanisms to survive in absence of MCU [18,19]. However, to best of our knowledge, not a single feedback module is reported for maintaining mitochondrial $Ca^{2+}$ homeostasis. Further, the functional significance of mitochondrial $Ca^{2+}$ dynamics in pigmentation biology remains undetermined.

Pigmentation is a complex physiological phenomenon that protects skin from UV induced DNA damage [20,21]. Inefficient pigmentation predisposes to skin cancers and perturbations in this pathway lead to pigmentary disorders such as vitiligo, melasma, Dowling Degos, etc. [20,21]. Pigmentation is an outcome of melanin synthesis (melanogenesis) in highly specialized lysosome-related organelles known as melanosomes. There are 4 stages of melanosomes wherein stages I-II are immature non-melanized while stages III-IV are highly melanized [20,21]. The role of mitochondria in regulating melanosome biology and thereby pigmentation has just started to emerge [22,23]. However, role of mitochondrial $Ca^{2+}$ uptake in melanosome biology remains completely unknown.

Here, we demonstrate a critical role of mitochondrial $Ca^{2+}$ uptake in vertebrate pigmentation. Using 2 independent in vitro (mouse B16 cells and primary human melanocytes) and 3 distinct in vivo (zebrafish, $MCU^{+/-}$ and $MCUb^{-/-}$ mice) models, we reveal that mitochondrial $Ca^{2+}$ uptake positively regulates melanogenesis. Mechanistically, decrease in mitochondrial $Ca^{2+}$ levels leads to activation and nuclear translocation of transcription factor NFAT2. NFAT2 in turn regulates transcription of keratin5, 7, and 8. These keratins enhance melanogenesis by augmenting melanosome biogenesis and maturation. Further, keratin5 amplifies mitochondrial $Ca^{2+}$ uptake thereby this signaling cascade functions as a feedback loop to maintain mitochondrial $Ca^{2+}$ homeostasis and to ensure optimum pigmentation. Finally, inhibition of MCU with an FDA approved drug mitoxantrone decreases physiological melanogenesis thereby highlighting potential of targeting mitochondrial $Ca^{2+}$ signaling to manage pigmentary disorders.

## Results

### Pigmentation is associated with enhanced mitochondrial $Ca^{2+}$ uptake

The functional significance of mitochondrial $Ca^{2+}$ signaling in pigmentation biology remains unappreciated. Therefore, we started by examining the mitochondrial $Ca^{2+}$ dynamics during

pigmentation. We measured mitochondrial $Ca^{2+}$ uptake in B16 mouse melanoma cells while they were synchronously pigmenting in a low-density (LD) culturing model. During LD pigmentation, non-pigmented B16 cells seeded at LD pigment over a period of 6 to 8 days [24,25] (**Fig 1A**). The LD pigmentation model closely recapitulates human melanogenic pathways [24–27]. We quantitated the increase in the melanogenesis during LD pigmentation model by performing melanin content assay, which showed a gradual increase in melanin synthesis from LD day 0 (D0) to LD day 4 (D4), LD day 5 (D5) and LD day 6 (D6) (**Fig 1B**). We used genetically encoded calcium-measuring organelle-entrapped protein indicators (CEPIA) for studying mitochondrial $Ca^{2+}$ dynamics [28]. We transfected B16 cells with a mitochondrial matrix specific CEPIA probe (CEPIA2mt) and temporally analyzed mitochondrial $Ca^{2+}$ uptake during LD pigmentation model. We used histamine, a physiological agonist, for studying mitochondrial $Ca^{2+}$ uptake. Histamine releases $Ca^{2+}$ from endoplasmic reticulum thereby enhancing cytosolic $Ca^{2+}$ levels. This rise in cytosolic $Ca^{2+}$ stimulates mitochondrial $Ca^{2+}$ uptake via MCU complex [29]. Our live cell mitochondrial $Ca^{2+}$ imaging assays demonstrate that with increase in pigmentation, the mitochondrial $Ca^{2+}$ uptake is enhanced (**Fig 1C and 1D**). The increase in mitochondrial $Ca^{2+}$ uptake could be independent of augmented expression and/or activity of MCU. This may happen when histamine stimulation results in enhanced cytosolic $Ca^{2+}$ levels during LD pigmentation. Therefore, we measured changes in cytosolic $Ca^{2+}$ levels upon histamine stimulation on LD D0, D4, D5, and D6. Our FURA-2AM based live cell $Ca^{2+}$ imaging reveals that histamine induced cytosolic $Ca^{2+}$ levels remain unchanged during LD pigmentation (**Fig 1E and 1F**). Secondly, we examined the overall mitochondrial content, using Mito-tracker staining, during LD pigmentation and found that the cellular mitochondrial content remains unchanged during pigmentation (**S1A and S1B Fig**). Thereby ruling out the possibility that the increased mitochondrial $Ca^{2+}$ uptake during pigmentation could be due to enhanced mitochondrial content. These crucial control experiment suggests that the increase in mitochondrial $Ca^{2+}$ uptake is most likely due to enhanced expression and/or activity of MCU. Hence, we analyzed changes in MCU expression during LD pigmentation and our western blotting data indicates that MCU expression is higher in pigmented LD D6 cells in comparison to non-pigmented LD D0 cells (**Fig 1G and 1H**). We further examined the levels of other MCU complex members (MCUb, EMRE, MICU1, and MICU2) during LD pigmentation and found that their expression is also enhanced at LD D6 (**S1C Fig**). Taken together, this data suggests that LD pigmentation is associated with enhanced mitochondrial $Ca^{2+}$ uptake and increased expression of MCU complex.

We next examined mitochondrial $Ca^{2+}$ levels in primary human melanocytes with varying pigment levels, i.e., lightly pigmented (LP) melanocytes of Caucasian origin and darkly pigmented (DP) melanocytes of Afro-American origin [23] (**Fig 1I**). We quantitated the differential pigmentation in LP and DP primary human melanocytes by performing melanin content assay, which showed robust increase in melanin synthesis in DP as compared to LP (**Fig 1J**). We used mitochondrial $Ca^{2+}$ measurement probe Rhod-2AM for investigating differences in mitochondrial $Ca^{2+}$ dynamics between primary LP and DP melanocytes. We observed that the mitochondrial $Ca^{2+}$ uptake is higher in Afro-American melanocytes in comparison to Caucasian melanocytes (**Fig 1K and 1L**). To exclude the possible artifacts associated with Rhod-2AM, we co-stained primary melanocytes with Mito-tracker green and normalized the Rhod-2AM signals with Mito-tracker green signals. This data set corroborate our observation that the mitochondrial $Ca^{2+}$ uptake is higher in DP melanocytes in comparison to LP melanocytes (**S1D Fig**). Further, we measured histamine stimulated changes in cytosolic $Ca^{2+}$ levels in LP and DP melanocytes. FURA-2AM based live cell $Ca^{2+}$ imaging shows that histamine induced cytosolic $Ca^{2+}$ levels are comparable in primary LP and DP melanocytes (**Fig 1M and 1N**). We next analyzed total mitochondrial content in LP and DP melanocytes and found the overall

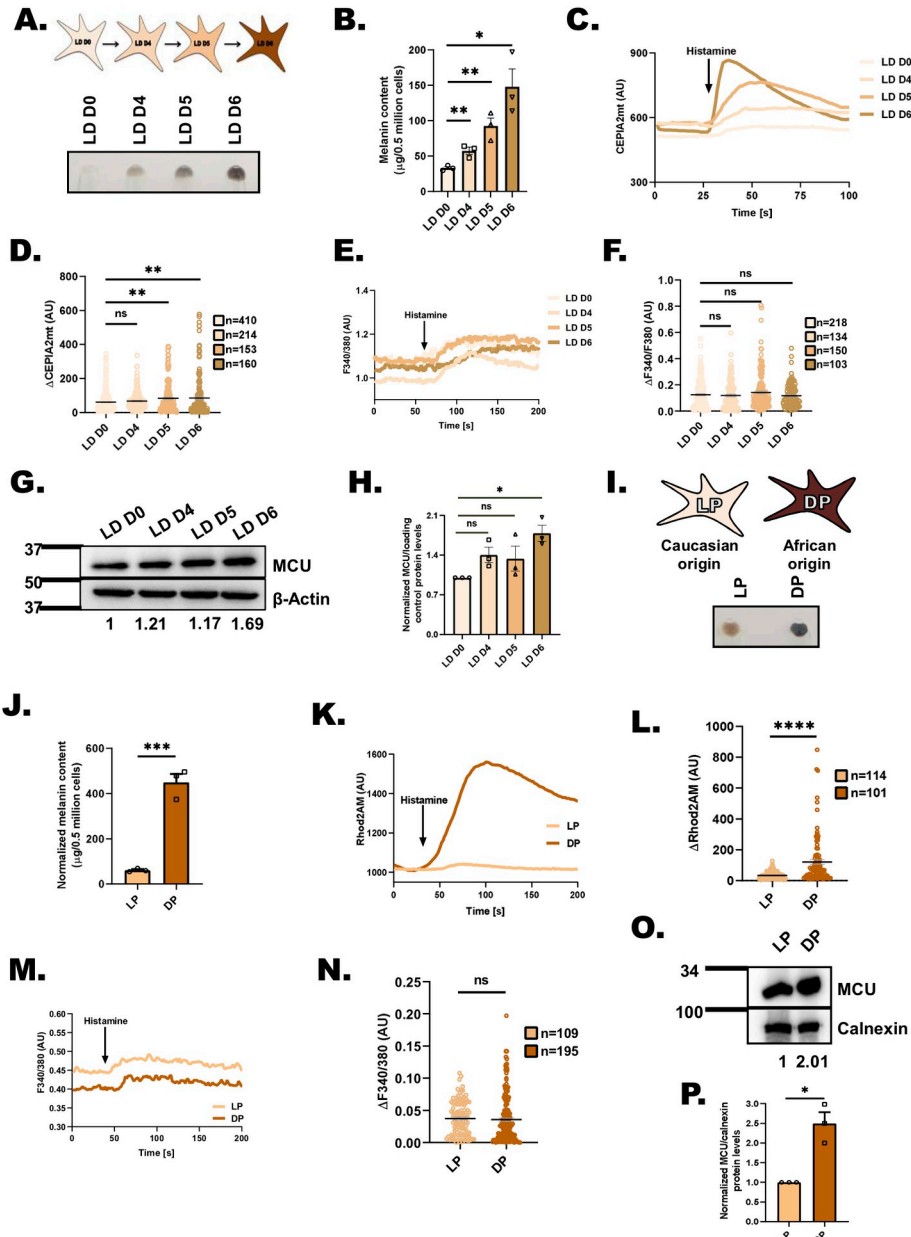

**Fig 1. Mitochondrial Ca²⁺ uptake is positively associated with melanogenesis.** **(A)** Representative B16 cell pellet images of LD day 0, LD day 4, LD day 5, and LD day 6 ($N = 3$). **(B)** Melanin content estimation of B16 cells on LD day 0, LD day 4, LD day 5, and LD day 6 ($N = 3$). **(C)** Representative mitochondrial Ca²⁺ imaging traces of CEPIA2mt on LD day 0, LD day 4, LD day 5, and LD day 6 B16 cells stimulated with 100 μm histamine. **(D)** Quantitation of mitochondrial Ca²⁺ uptake by calculating ΔCEPIA2mt on LD day 0, LD day 4, LD day 5, and LD day 6 B16 cells stimulated with 100 μm histamine where "$n$" denotes the number of ROIs. **(E)** Representative traces of Fura-2 imaging to measure cytosolic Ca²⁺ on LD day 0, LD day 4, LD day 5, and LD day 6 B16 cells stimulated with 100 μm histamine. **(F)** Quantitation of cytosolic Ca²⁺ levels on LD day 0, LD day 4, LD day 5, and LD day 6 B16 cells stimulated with 100 μm histamine where "$n$" denotes the number of ROIs. **(G)** Representative western blot showing expression of MCU on LD day 0, LD day 4, LD day 5, and LD day 6. Densitometric analysis using ImageJ is presented below the blot ($N = 3$). **(H)** Densitometric quantitation showing MCU levels on LD day 0, LD day 4, LD day 5, and LD day 6 ($N = 3$). **(I)** Representative pellet images of LP and DP primary human melanocytes ($N = 3$). **(J)** Melanin content estimation of LP and DP primary human melanocytes ($N = 3$). **(K)** Representative mitochondrial Ca²⁺ imaging traces of LP and DP primary human melanocytes stimulated with 100 μm histamine. **(L)** Quantitation of ΔRhod-2 in LP and DP primary human melanocytes stimulated with 100 μm histamine where "$n$" denotes the number of ROIs. **(M)** Representative traces of Fura-2 imaging to measure cytosolic Ca²⁺ in LP and DP primary human melanocytes stimulated with 100 μm histamine. **(N)** Quantitation of cytosolic Ca²⁺ levels in LP and DP primary human melanocytes stimulated with

100 μm histamine where "*n*" denotes the number of ROIs. **(O)** Representative western blot showing an increase in MCU protein expression in DP primary human melanocytes in comparison to LP primary human melanocytes. Densitometric analysis using ImageJ is presented below the blot ($N = 3$). **(P)** Densitometric quantitation showing increase in MCU protein levels in DP primary human melanocytes in comparison to LP primary human melanocytes ($N = 3$). Data presented are mean ± SEM. For statistical analysis, unpaired Student's *t* test was performed for panels B, J, L, and N, one sample *t* test was performed for panel P while one-way ANOVA followed by Tukey's post hoc test was performed for panel D, F, and H using GraphPad Prism software. Here, ns means nonsignificant; * $p < 0.05$; ** $p < 0.01$; *** $p < 0.001$; and **** $p < 0.0001$. The data underlying for panels B, C, D, E, F, H, J, K, L, M, N, and P shown in the figure can be found in S1 Data. DP, darkly pigmented; LD, low density; LP, lightly pigmented; MCU, mitochondrial calcium uniporter.

mitochondrial content is similar in primary melanocytes (**S1E and S1F Fig**). We then examined MCU expression in LP and DP melanocytes and found that the levels of MCU protein are about 2-fold higher in DP melanocytes in comparison to LP melanocytes (**Fig 1O and 1P**). Similarly, the expression of other MCU complex members (MCUb, EMRE, MICU1, and MICU2) is higher in DP melanocytes as compared to LP melanocytes (**S1G Fig**). This suggests that most likely augmented levels of MCU complex contribute to enhanced mitochondrial $Ca^{2+}$ uptake observed in DP melanocytes. Altogether, data from both B16 pigmentation model and primary human melanocytes highlight that increased pigmentation is associated with higher mitochondrial $Ca^{2+}$ uptake and enhanced expression of MCU complex. However, the functional relevance of MCU in pigmentation biology remains completely unknown. Therefore, we investigated its role in pigmentation.

## MCU positively regulates melanogenesis while MCUb negatively control melanogenesis

To understand the role of MCU in melanogenesis, we performed loss of function and gain of function studies in B16 cells. We silenced MCU in B16 cells (**Fig 2A**) and observed a drastic decrease in MCU mRNA levels in siMCU cells in comparison to control non-targeting siRNA (siNT) transfected cells (**S2A Fig**). We validated MCU knockdown by performing western blotting. We observed around 70% decrease in the MCU protein expression in siMCU transfected cells (**Figs 2B** and **S2B**). Next, we examined the effect of MCU silencing on B16 mitochondrial $Ca^{2+}$ dynamics. Using CEPIA2mt, we measured resting mitochondrial $Ca^{2+}$ levels as well as histamine and α-melanocyte stimulating hormone (αMSH, a physiological melanogenic stimuli that can mediate ER $Ca^{2+}$ release) induced mitochondrial $Ca^{2+}$ uptake upon MCU silencing. In these live cell imaging experiments, we observed that MCU silencing decreases resting mitochondrial $Ca^{2+}$ levels (**Figs 2C, 2D, S2C, and S2D**). Further, we saw a significant reduction in histamine (**Fig 2C and 2E**) and αMSH (**Fig 2D and 2F**) stimulated mitochondrial $Ca^{2+}$ uptake upon MCU knockdown (**Fig 2C–2F**). We then assessed changes in cytosolic $Ca^{2+}$ levels upon MCU silencing and found that both histamine and αMSH-mediated cytosolic $Ca^{2+}$ levels were increased in MCU knockdown cells in comparison to control cells (**S2E–S2H Fig**). Taken together, these results suggest that MCU silencing in B16 cells decreases both resting mitochondrial $Ca^{2+}$ levels and agonist induced mitochondrial $Ca^{2+}$ uptake while the cytosolic $Ca^{2+}$ levels are increased.

We then investigated role of MCU in B16 LD melanogenesis. We performed siRNA-mediated knockdown of MCU on LD day 3 in B16 LD pigmentation model and examined LD cells on day 6. The knockdown of MCU resulted in visible decrease in LD day 6 pigmentation as evident in the LD day 6 pellet images (**Fig 2G**). We further quantitated the decrease in melanogenesis upon MCU silencing by performing melanin content assays and observed a significant decrease in melanogenesis (**Fig 2H**). To further corroborate MCU's role in melanogenesis, we performed gain of function studies. For these experiments, we used non-pigmented B16 cells

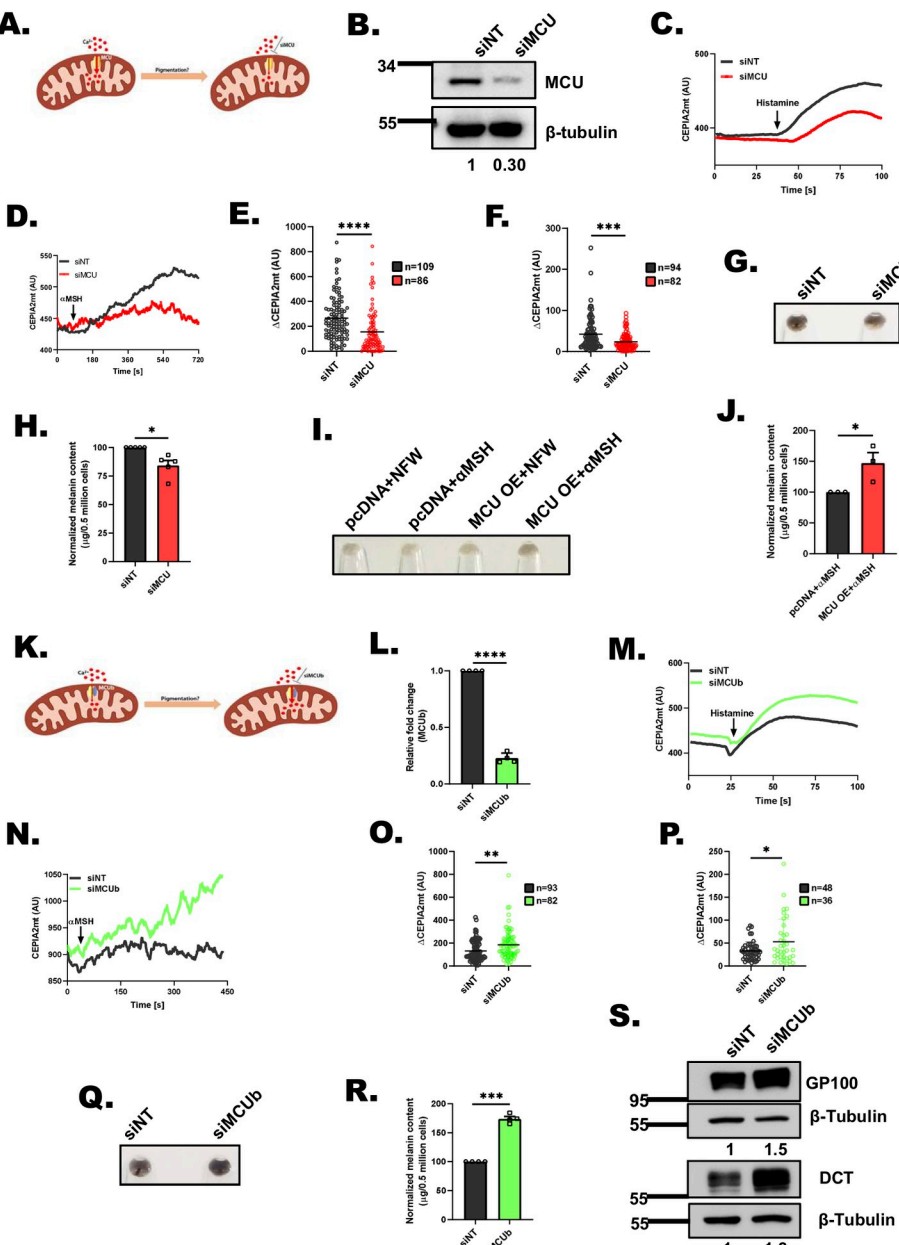

**Fig 2. MCU positively regulates melanogenesis while MCUb negatively controls melanogenesis. (A)** Schematic showing effect of MCU silencing on mitochondrial $Ca^{2+}$ uptake. **(B)** Representative western blot confirming siRNA based silencing of MCU on LD day 6 B16 cells. Densitometric analysis using ImageJ is presented below the blot ($N = 3$). **(C)** Representative mitochondrial $Ca^{2+}$ imaging traces of CEPIA2mt in siNon-Targeting (siNT) control and siMCU B16 cells stimulated with 100 μm histamine. **(D)** Representative mitochondrial $Ca^{2+}$ imaging traces of CEPIA2mt in siNon-Targeting (siNT) control and siMCU B16 cells stimulated with 1 μm αMSH. **(E)** Quantitation of mitochondrial $Ca^{2+}$ uptake by calculating increase in CEPIA2mt signal (ΔCEPIA2mt) in siNT control and siMCU B16 cells stimulated with 100 μm histamine where "$n$" denotes the number of ROIs. **(F)** Quantitation of mitochondrial $Ca^{2+}$ uptake by calculating increase in CEPIA2mt signal (ΔCEPIA2mt) in siNT control and siMCU B16 cells stimulated with 1 μm αMSH where "$n$" denotes the number of ROIs. **(G)** Representative pellet images of siNT control and siMCU on LD day 6 ($N = 5$). **(H)** Melanin content estimation of siNT and siMCU B16 cells on LD day 6 ($N = 5$). **(I)** Representative pellet images of pcDNA control plasmid and MCU-GFP overexpression either treated with NFW or αMSH ($N = 3$). **(J)** Melanin content estimation of pcDNA control plasmid and MCU-GFP overexpression upon αMSH treatment ($N = 3$). **(K)** Schematic showing effect of MCUb silencing on mitochondrial $Ca^{2+}$ uptake. **(L)** qRT-PCR analysis showing decrease in MCUb mRNA expression upon MCUb silencing in B16 cells ($N = 4$). **(M)** Representative mitochondrial $Ca^{2+}$ imaging traces of CEPIA2mt in siNT control and siMCUb B16 cells stimulated

with 100 μm histamine. **(N)** Representative mitochondrial $Ca^{2+}$ imaging traces of CEPIA2mt in siNT control and siMCUb B16 cells stimulated with 1 μm αMSH. **(O)** Quantitation of mitochondrial $Ca^{2+}$ uptake by calculating increase in CEPIA2mt signal (ΔCEPIA2mt) in siNT control and siMCUb B16 cells upon stimulation with 100 μm histamine where "$n$" denotes the number of ROIs. **(P)** Quantitation of mitochondrial $Ca^{2+}$ uptake by calculating increase in CEPIA2mt signal (ΔCEPIA2mt) in siNT control and siMCUb B16 cells upon stimulation with 1 μm αMSH where "$n$" denotes the number of ROIs. **(Q)** Representative pellet images of siNT control and siMCUb on LD day 6 ($N = 4$). **(R)** Melanin content estimation in siNT and siMCUb B16 cells on LD day 6 ($N = 4$). **(S)** Representative western blot showing expression of GP100 and DCT on LD day 6 upon MCUb silencing as compared to siNT control. Densitometric analysis using ImageJ is presented below the blot ($N = 3$). Data presented are mean ± SEM. For statistical analysis, unpaired Student's $t$ test was performed for panels E, F, J, O, and P, while one sample $t$ test was performed for panels H, L, and R using GraphPad Prism software. Here, $^*$ $p < 0.05$; $^{**}$ $p < 0.01$; $^{***}$ $p < 0.001$; and $^{****}$ $p < 0.0001$. The data underlying for panels C, D, E, F, H, J, L, M, N, O, P, and R shown in the figure can be found in S1 Data. LD, low density; MCU, mitochondrial calcium uniporter; NFW, nuclease free water.

cultured in high density (HD) and induced melanogenesis with αMSH. We used αMSH as a stimuli for stimulating melanogenesis in B16 HD cells as it induces mitochondrial $Ca^{2+}$ uptake. We overexpressed MCU in B16 cells by transfecting human MCU-GFP plasmid (**S2I Fig**) and confirmed MCU overexpression (OE) by performing western blotting. As MCU was tagged with GFP in the overexpression construct, we observed a higher molecular-weight band corresponding to MCU-GFP (**S2J Fig**). We first confirmed that MCU overexpression results in an increase in histamine and αMSH induced mitochondrial $Ca^{2+}$ uptake (**S2K–S2N Fig**) as well as corresponding decrease in cytosolic $Ca^{2+}$ levels (**S3A–S3C Fig**). We then analyzed the levels of other MCU complex members (EMRE, MCUb MICU1, and MICU2) upon MCU overexpression and found that their expression remains unchanged upon ectopic expression of MCU (**S3D Fig**). This suggests that in our system increase in MCU expression is enough to enhance mitochondrial $Ca^{2+}$ uptake. Next, we examined the effect of MCU overexpression on αMSH-stimulated pigmentation and noticed that MCU overexpression increases αMSH-induced pigmentation. The pellet images from pcDNA (empty vector control)+NFW (nuclease free water; vehicle control for αMSH), pcDNA+αMSH, MCU-GFP OE+NFW, and MCU-GFP OE+αMSH conditions clearly show increased pigmentation in MCU-GFP OE+αMSH condition (**Fig 2I**). We further quantitated these phenotypic changes by performing melanin-content assays and observed that the ectopic expression of MCU-GFP leads to a significant increase in αMSH-induced melanogenesis in comparison to empty vector control (**Fig 2J**). Taken together, our results demonstrate that MCU positively regulates melanogenesis in B16 cells.

To further corroborate role of MCU in pigmentation, we studied it in primary human melanocytes. To examine the role of MCU in regulating melanogenesis, we silenced MCU using human siRNAs in LP melanocytes. We validated MCU knockdown by performing qRT-PCR and western blot analysis. We observed around 70% decrease in MCU mRNA levels (**S3E Fig**) and over 50% decrease in MCU protein expression in primary melanocytes transfected with siMCU as compared to siNT (**S3F and S3G Fig**). The knockdown of MCU resulted in visible decrease in melanogenesis as evident in the pellet images of primary melanocytes (**S3H Fig**). We quantitated the change in melanogenesis upon MCU silencing in primary melanocytes by Image J based analysis and observed a significant reduction in pigmentation in siMCU condition in comparison siNT control (**S3I Fig**). To further strengthen this phenotypic data, we analyzed mRNA expression of key melanogenic enzymes, i.e., tyrosinase, tyrosinase-related protein 1 (TYRP1), and tyrosinase-related protein 2/Dopachrome Tautomerase (DCT) in siNT and siMCU cells. We observed that MCU silencing results in significant reduction in tyrosinase, TYRP1 and DCT expression (**S3J Fig**). Collectively, the experiments in 3 independent melanogenesis models (B16 LD model, B16 αMSH-induced pigmentation and melanogenesis in primary melanocytes) clearly demonstrate a critical role of MCU in driving melanogenesis.

Next, we questioned whether the changes in melanogenesis upon altering MCU expression is due to concomitant fluctuations in mitochondrial $Ca^{2+}$ levels or they are associated with some $Ca^{2+}$ independent functions of MCU. To address this, we silenced MCUb, an important negative regulator of MCU [30] in B16 LD pigmentation model (**Fig 2K**). We first characterized MCUb siRNA and observed a significant reduction in MCUb mRNA levels in siMCUb transfected cells in comparison to control siNT condition (**Fig 2L**). Next, we measured basal mitochondrial $Ca^{2+}$ levels as well as histamine and αMSH induced mitochondrial $Ca^{2+}$ uptake upon MCUb silencing. We noticed a significant increase in basal mitochondrial $Ca^{2+}$ levels (**Figs 2M, 2N, S4A and S4B**) as well as histamine and αMSH stimulated mitochondrial $Ca^{2+}$ uptake upon MCUb knockdown (**Fig 2M–2P**). This data demonstrates that MCUb silencing enhances mitochondrial $Ca^{2+}$ uptake. We next assessed changes in cytosolic $Ca^{2+}$ levels upon MCUb silencing and found that both histamine and αMSH mediated cytosolic $Ca^{2+}$ levels were decreased in MCUb knockdown condition in comparison to control cells (**S4C–S4F Fig**). We then evaluated role of MCUb in B16 LD pigmentation model. We observed a visible increase in LD day6 pigmentation upon knockdown of MCUb (**Fig 2Q**). We quantitated the increase in melanogenesis by performing melanin content assays and noted approximately 75% augmentation in melanogenesis upon MCUb silencing (**Fig 2R**). We next examined the expression of key melanosome structural protein (Pre-melanosome Protein 17, i.e., PMEL17 or GP100) and melanogenic enzymes (tyrosinase and DCT) upon the silencing of MCUb. The siMCUb cells expressed significantly higher levels of GP100 and DCT as compared to siNT control cells (**Figs 2S, S4G and S4H**). Further, we investigated expression and activity of the rate-limiting enzyme in melanin synthesis pathway, i.e., tyrosinase. We performed DOPA (dopachrome generation) assay for examining tyrosinase activity. Tyrosinase converts its substrate L-DOPA to Dopachrome that in turn yields a brown-to-black color on native gels. We observed that both the expression and activity of tyrosinase was significantly increased in siMCUb cells in comparison to siNT cells (**S4I–S4K Fig**). This data shows that MCUb is a negative regulator of melanogenesis.

We next corroborated the role of MCUb in regulating melanogenesis in primary human melanocytes. We transiently silenced MCUb using siRNAs in LP primary human melanocytes. We first validated MCUb knockdown in primary cells and observed around 60% decrease in the MCUb mRNA levels in siMCUb transfected primary melanocytes (**S4L Fig**). The knockdown of MCUb resulted in the visible increase in melanogenesis (**S4M Fig**). Further, we analyzed mRNA expression of key melanogenic enzymes and melanosome structural proteins, i.e., GP100, tyrosinase and DCT in the siNT and siMCUb cells. We observed that MCUb silencing results in the significant increase in the expression of GP100, tyrosinase, and DCT (**S4N Fig**). These findings suggest that the increase in expression of key melanogenic enzymes and melanosome structural protein contribute to enhanced melanogenesis observed in MCUb silenced melanocytes. Taken together, the data from B16 LD model and primary melanocytes establish MCUb as a negative regulator of pigmentation. Collectively, the data presented in **Figs 2 and S2–S4** demonstrate that mitochondrial $Ca^{2+}$ uptake positively regulates pigmentation.

Next, we investigated role of MCU and MCUb in regulating other mitochondrial functions in B16 cells. We performed Seahorse analysis to assess the mitochondrial respiration and glycolysis upon MCU and MCUb overexpression in B16 cells. Our data suggests that MCU does not play a significant role in modulating mitochondrial respiration and glycolysis while MCUb positively regulates basal mitochondrial respiration (**S5A–S5I Fig**). We then examined changes in mitochondrial membrane potential in MCU and MCUb silenced B16 cells. We carried out flow cytometry with TMRE stain and observed that MCU and MCUb do not modulate mitochondrial membrane potential (**S5J Fig**). Finally, we measured ATP production in

B16 cells using luminescence-based ATP assays. Our data shows that MCU and MCUb do not significantly contribute to ATP production in B16 cells (**S5K Fig**). The luminescence-based ATP measurements matches with the Seahorse analysis wherein we do not observe changes in ATP production. Overall, this mitochondrial function analysis is largely in-line with the published literature wherein MCU and MCUb have been reported to play a nonsignificant role in regulating mitochondrial respiration and mitochondrial membrane potential [11,12,30,31].

## MCU and MCUb regulates pigmentation in vivo

Using zebrafish as a model system, we next investigated role of MCU in pigmentation in vivo. Zebrafish are extensively used as a model system for pigmentation studies [32] as signaling events and molecular players involved in regulating pigmentation are largely conserved across vertebrates. Zebrafish embryos are transparent and the pigmented melanophores, which are melanocyte equivalents in zebrafish appear within 36 to 48 hours post fertilization (hpf). The zebrafish pigmentation can be quantified with microscopic analysis and biochemical assays [33]. Thus, zebrafish serves as an exciting model system to examine the relevance of pigmentation regulators in vivo.

In first set of experiments, we employed morpholino (MO) based knockdown strategy. We used specific morpholinos (injected at single-cell stage) targeting zebrafish MCU and then followed changes in pigmentation. We characterized the morpholinos targeting MCU by performing western blotting on zebrafish lysates 48 hpf. We observed around 60% decrease in the MCU protein expression upon MCU morpholino injection as compared to control morpholino (**Figs 3A** and **S6A**). Phenotypically, we observed a substantial reduction in the pigmented black-colored melanophores in MCU morphants in comparison to the control embryos at 30 hpf and 48 hpf (**Fig 3B and 3C**). The decreased pigmentation phenotype was observed in 70% embryos at 30 hpf (**S6B Fig**). Next, we quantitated the reduction in melanogenesis by performing melanin content assays and found that the melanin content was decreased by approximately 60% in MCU knockdown zebrafish embryos in comparison to control embryos (**Fig 3D**). To further corroborate in vivo data, we performed gain of function and rescue studies in zebrafish. We injected human MCU RNA in zebrafish and observed increase in the pigmented melanophores as compared to the control embryos at 48 hpf (**S6C Fig**). We quantitated the increase in pigmentation by performing melanin content assays and detected approximately 40% increase in melanin content upon human MCU RNA injections in zebrafish (**S6D Fig**). Moreover, we performed rescue experiments by injecting human MCU RNA along with MCU morpholinos. We observed rescue of hypopigmentation phenotype upon injection of human MCU RNA (**Fig 3E**) in MCU morphant embryos. The quantitative melanin content analysis on the zebrafish embryos clearly demonstrates that the decrease in pigmentation observed upon MCU knockdown in zebrafish can be rescued by co-injection of human MCU RNA (**Fig 3F**). Taken together, our zebrafish data elegantly establish an important role of MCU in regulating pigmentation in vivo.

Next, we assessed role of MCU and MCUb in mice epidermal pigmentation. For addressing this question, we utilized 2 recently reported mice models, i.e., global MCU heterozygous knockout (MCU$^{+/-}$) and complete MCUb knockout (MCUb$^{-/-}$) mice [31,34]. We studied epidermal pigmentation in MCU$^{+/-}$ mice instead of MCU knockout (MCU$^{-/-}$) mice as viable MCU$^{-/-}$ mice are reported to show compensation and therefore many times do not display phenotypic changes [18,19]. First of all, we validated the effect of MCU heterozygous and MCUb global knockout on mitochondrial Ca$^{2+}$ uptake in skin fibroblasts isolated from these transgenic mice and corresponding wild-type control mice. As expected, we observed a significant decrease in both resting mitochondrial Ca$^{2+}$ levels and mitochondrial Ca$^{2+}$ uptake in

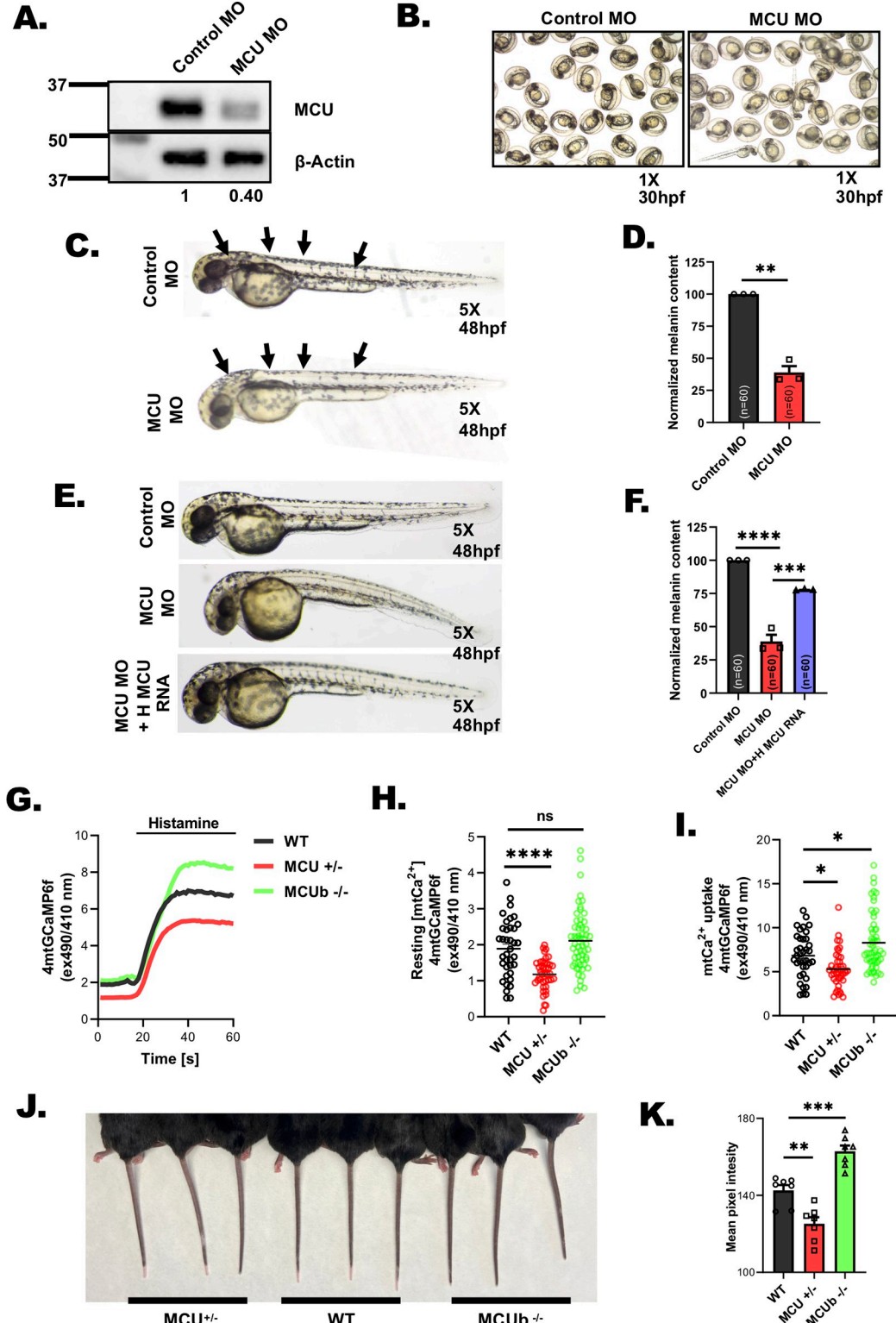

**Fig 3. MCU regulates pigmentation in vivo. (A)** Representative western blot showing expression of MCU in control MO and MCU MO. Densitometric analysis using ImageJ is presented below the blot ($N$ = 3). **(B)** Representative bright-field images of wild-type zebrafish embryos injected with either control morpholino or morpholino targeting zebrafish MCU at 30 hpf ($N$ = 3 independent experiments with approximately 200 embryos/condition). **(C)** Representative bright-field images of wild-type zebrafish embryos injected with either control morpholino or morpholino targeting zebrafish MCU at

48 hpf ($N$ = 3 independent experiments with approximately 200 embryos/condition). **(D)** Melanin-content estimation of control MO and MCU MO in zebrafish embryos in around 60 embryos from 3 independent sets of injections ($N$ = 3 independent experiments with 60 embryos/condition). **(E)** Representative bright-field images of zebrafish embryos injected with control morpholino; morpholino targeting zebrafish MCU and MCU morpholino injected with human MCU RNA ($N$ = 3 independent experiments with approximately 200 embryos/condition). **(F)** Melanin-content estimation of control MO; MCU MO and MCU MO injected with human MCU RNA in zebrafish embryos in around 60 zebrafish embryos from 3 independent sets of injections ($N$ = 3 independent experiments with 60 embryos/condition). **(G)** Representative mitochondrial $Ca^{2+}$ imaging traces of skin fibroblasts isolated from wild-type and $MCU^{+/-}$ and $MCUb^{-/-}$ mice stimulated with 100 μm histamine. **(H)** Quantitation of resting mitochondrial $Ca^{2+}$ in skin fibroblasts isolated from wild-type and $MCU^{+/-}$ and $MCUb^{-/-}$ mice. **(I)** Quantitation of mitochondrial $Ca^{2+}$ uptake in skin fibroblasts isolated from wild-type and $MCU^{+/-}$ and $MCUb^{-/-}$ mice stimulated with 100 μm histamine. **(J)** Representative images of tail from 8- to 12-week-old wild-type and $MCU^{+/-}$ and $MCUb^{-/-}$ mice. **(K)** Quantitation of mean pixel intensity of tail pigmentation measured by ImageJ software in wild-type mice and $MCU^{+/-}$ and $MCUb^{-/-}$ mice ($N$ = 7 data points from 7 independent mice/condition). Data presented are mean ± SEM. For statistical analysis, one sample $t$ test was performed for panel D while one-way ANOVA followed by Tukey's post hoc test was performed for panel F, H, I, and K using GraphPad Prism software. Here, ** $p < 0.01$; *** $p < 0.001$; and **** $p < 0.0001$. The data underlying for panels D, F, G, H, I, and K shown in the figure can be found in S1 Data. hpf, hours post fertilization; MCU, mitochondrial calcium uniporter.

$MCU^{+/-}$ mice cells as compared to cells harvested from wild-type mice (**Fig 3G–3I**). Likewise, the calcium imaging experiments in cells isolated from $MCUb^{-/-}$ mice show that there is a significant increase in mitochondrial $Ca^{2+}$ uptake while resting mitochondrial $Ca^{2+}$ levels are nonsignificantly augmented in comparison to cells harvested from wild-type mice (**Fig 3G–3I**). After confirming the anticipated changes in the mitochondrial $Ca^{2+}$ signaling in these mice, we examined the epidermal pigmentation in tail of these transgenic mice. Excitingly, we observed reduced tail pigmentation in $MCU^{+/-}$ mice, whereas $MCUb^{-/-}$ mice showed enhanced pigmentation in tail in comparison to control mice (**Fig 3J**). It is important to note that epidermal pigmentation in mice is typically studied in mice tail as mice fur pigmentation is equivalent to human hair pigmentation. We further quantitated the differences in epidermal pigmentation between wild-type and $MCU^{+/-}$ as well as wild-type and $MCUb^{-/-}$ mice with a standard protocol used for estimating pigmentation in animal models [27]. Briefly, we calculated mean pixel intensity of the tail images using ImageJ software. Notably, we observed a significant decrease in the epidermal tail pigmentation in $MCU^{+/-}$ mice while tail pigmentation was significantly increased in $MCUb^{-/-}$ mice (**Fig 3K**). This data further validates that MCU complex plays a critical role in pigmentation in vivo. Collectively, our data from 4 independent model systems, i.e., (1) B16 cells; (2) primary human melanocytes; (3) Zebrafish; and (4) 2 distinct transgenic mice models clearly establish that mitochondrial $Ca^{2+}$ uptake plays a crucial role in driving vertebrate pigmentation.

## Unbiased transcriptome analysis identifies keratin filaments as key melanogenic regulators working downstream of mitochondrial $Ca^{2+}$ signaling

To identify the molecular mechanism regulating pigmentation downstream of mitochondrial $Ca^{2+}$, we performed transcriptome profiling (RNA sequencing). RNA seq was performed on siNT, siMCU, and siMCUb LD day 6 cells, (**Fig 4A**) using Illumina NovaSeq. The data quality was checked using FastQC and MultiQC software. The QC passed reads were mapped onto indexed Mus musculus genome (GRCm39) using STAR v2 aligner. Gene level expression values were obtained as read counts using feature-counts software. Next, we carried out differential expression analysis using the DESEq2 package. The fold change was calculated for siMCU and siMCUb samples with respect to siNT control. Genes showing expression more than log2 fold change +1 with respect to siNT were taken as up-regulated genes in siMCU and siMCUb. While a cut off of log2 fold change less than −1 was set for determining down-regulated genes.

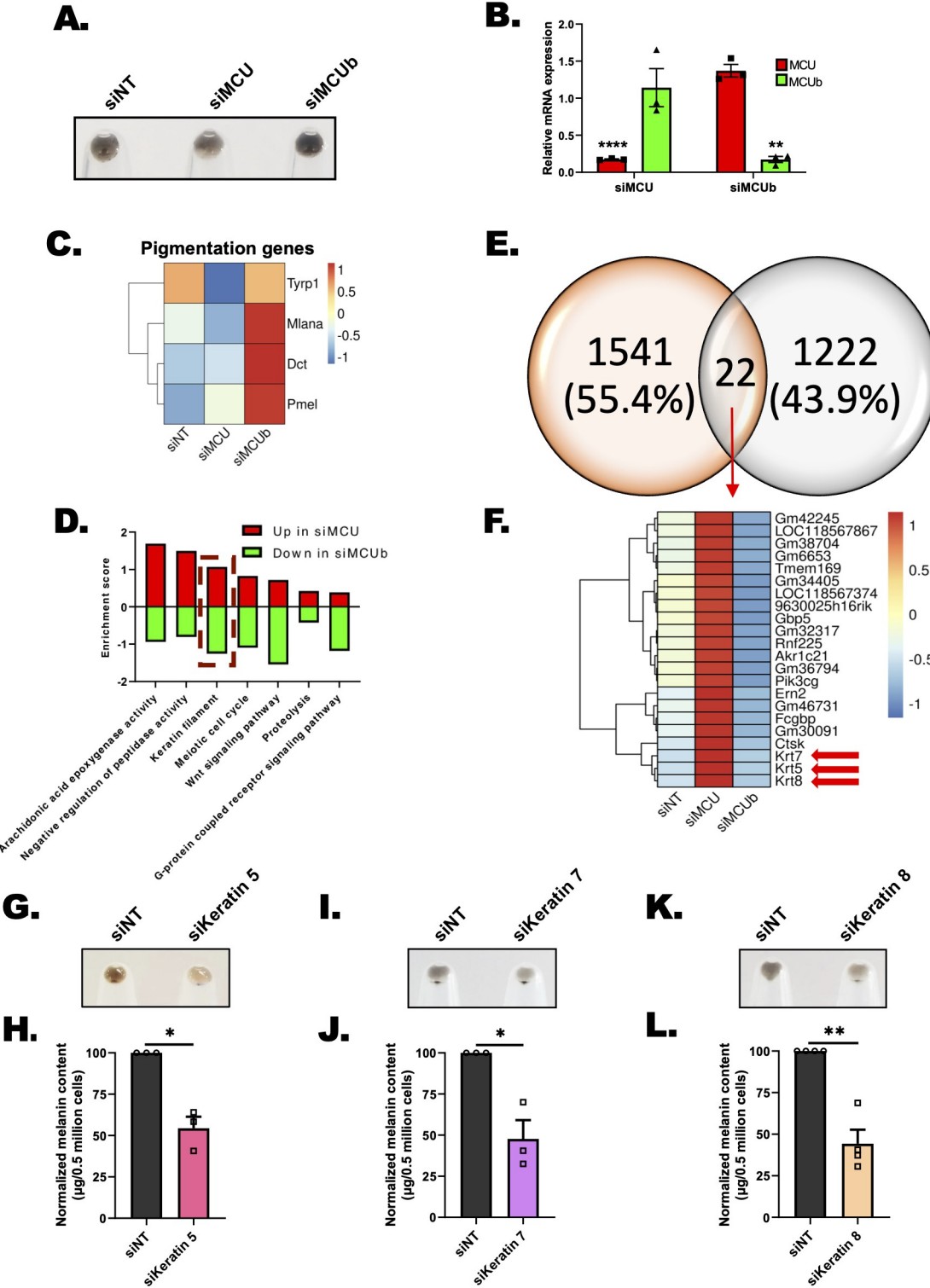

**Fig 4. Transcriptomics identifies keratin filaments working downstream of mitochondrial Ca²⁺ dynamics to regulate melanogenesis. (A)** Representative pellet images of B16 LD cells used for RNA sequencing. **(B)** qRT-PCR analysis showing decrease in MCU and MCUb mRNA expression upon MCU and MCUb silencing, respectively ($N$ = 3). **(C)** Heatmap representing the expression of pigmentation genes upon silencing of MCU and MCUb, respectively. Scale from blue to red represents z-score for fold change from −1 to +1. **(D)** Common oppositely regulated pathways up in siMCU and down in siMCUb. **(E)** Venn Diagram representing common differentially regulated genes upon silencing of MCU and MCUb. (F) Heatmap representing the

expression of 22 common differentially regulated genes upon silencing of MCU and MCUb, respectively. Scale from blue to red represents z-score for fold change from −1 to +1. (G) Representative pellet images of siNT control and siKeratin 5 on LD day 6 ($N$ = 3). (H) Melanin content estimation of siNT and siKeratin 5 B16 cells on LD day 6 ($N$ = 3). (I) Representative pellet images of siNT control and siKeratin 7 on LD day 6 ($N$ = 3). (J) Melanin content estimation of siNT and siKeratin 7 B16 cells on LD day 6 ($N$ = 3). (K) Representative pellet images of siNT control and siKeratin 8 on LD day 6 ($N$ = 4). (L) Melanin content estimation of siNT and siKeratin 8 B16 cells on LD day 6 ($N$ = 4). Data presented are mean ± SEM. For statistical analysis, one sample $t$ test was performed for panels B, H, J, L using GraphPad Prism software. Here, * $p < 0.05$; ** $p < 0.01$; and **** $p < 0.0001$. The data underlying for panels B, C, D, H, J, and L shown in the figure can be found in S1 Data. LD, low density; MCU, mitochondrial calcium uniporter.

We first examined the levels of MCU and MCUb in the RNA seq data and observed a significant decrease in their levels in the respective siRNA condition (S7A Fig). We then validated knockdown of MCU and MCUb in RNA seq samples by performing qRT-PCR analysis (Fig 4B). As a positive control, we checked expression of pigmentation genes upon knockdown of MCU and MCUb in RNA seq samples. We observed increase in expression of pigmentation genes upon knockdown of MCUb while MCU silencing led to decrease in expression of some of the key pigmentation genes (Fig 4C). Taken together, this initial scrutiny gave us confidence that our RNA sequencing is performed and analyzed appropriately.

To identify the pathways that are modulated upon MCU and MCUb silencing, we performed pathway enrichment analysis on up-regulated and down-regulated genes (in siMCU and siMCUb conditions), using DAVID software. Pathway enrichment analysis was performed based on the Gene Ontology (GO) database and enriched pathways were plotted with enrichment scores. Since we observed opposing pigmentation phenotypes upon MCU and MCUb silencing, we adopted the strategy of determining the pathways that are oppositely regulated in siMCU and siMCUb conditions. We plotted the common but oppositely modulated pathways, i.e., up in siMCU and down in siMCUb and vice versa (down in siMCU and up in siMCUb) (Figs 4D and S7B). Top oppositely regulated pathways up in siMCU and down in siMCUb were arachidonic acid epoxygenase activity, keratin filaments, Wnt signaling, and G-protein coupled receptor signaling pathway among others. Likewise, top common but oppositely regulated pathways down in siMCU and up in siMCUb were integral component of membrane, ion transport, oxidoreductase activity, and chloride channel activity.

Next, to identify common but oppositely regulated genes, we plotted Venn diagram for genes up in siMCU and down in siMCUb. We observed 22 such differentially regulated genes upon silencing of MCU and MCUb (Fig 4E). Interestingly 3 out of 22 differentially regulated genes belong to keratin intermediate filament family, i.e., keratin5, 7, and 8 (Fig 4F). Upon detailed literature survey we found that the role of keratins in regulating melanogenesis remains poorly understood. Although mutations in keratin5 are associated with human hyper-pigmentary skin disorders such as Dowling Degos disease and epidermolysis bullosa simplex with mottled pigmentation [35,36], the underlying mechanism remains unexplained. Interestingly, a recent study demonstrated that the expression of keratin5 is higher in hyper-pigmentary conditions such as senile lentigo, photo-exposed skin, and Dowling Degos disease in comparison to normal mammary skin [37]. The authors further reported that keratin5 levels are lower in depigmented vitiligo skin [37]. These clinical studies show an association between keratin5 mutations/expression profile with pigmentary disorders. But whether the alteration in keratin5 expression is a cause or consequence of the disorders remains unknown. Moreover, the signaling cascades that drive keratin5 expression during pigmentation are not understood. Therefore, based on pathway enrichment analysis and literature survey we focused on understanding the role of keratin filaments in regulating melanogenesis downstream of mitochondrial $Ca^{2+}$ signaling.

## Keratin5 regulates mitochondrial $Ca^{2+}$ uptake, melanosome maturation, and positioning

To understand the role of keratin5, 7, and 8 in pigmentation, we characterized the siRNAs targeting them. As presented in S7C–S7E Fig, siRNAs against keratin5, 7, and 8 decrease the target mRNAs by approximately 75% in B16 cells. The knockdown of all 3 keratins led to a substantial phenotypic decrease in B16 LD pigmentation as evident in the LD day 6 pellet images (Fig 4G, 4I, and 4K). We further quantitated the decrease in pigmentation by performing melanin content assays and observed that silencing of keratin5, 7, and 8 decreased melanogenesis by over 50% (Fig 4H, 4J, and 4L). Hence, our data suggests that keratin5, 7, and 8 positively regulate B16 LD melanogenesis. Since all 3 keratins (keratin5, 7, and 8) gave similar pigmentation phenotype and clinically keratin5 is associated with pigmentary disorders, we focused on keratin5 for subsequent studies. We first validated role of keratin5 in regulating pigmentation in primary human melanocytes. We silenced keratin5 in primary human melanocytes using human keratin5 siRNA and observed around 80% decrease in keratin5 mRNA levels (S7F Fig). The knockdown of keratin5 resulted in visible decrease in melanogenesis as evident in the pellet images (S7G Fig). These observations were rather surprising as the expression of keratin5 increases upon MCU silencing and decreases upon MCUb knockdown (Fig 4D and 4F). Therefore, we were expecting that silencing of keratin5 would result in increase in melanogenesis as observed in case of siMCUb. Our pigmentation data suggested that keratin5 could be playing a compensatory/feedback function downstream of mitochondrial $Ca^{2+}$ signaling. Recent literature suggests that keratin filaments can modulate mitochondrial organization and function [38–40]. However, role of keratins in regulating mitochondrial $Ca^{2+}$ signaling remains unappreciated. Therefore, we investigated role of keratin5 in controlling mitochondrial $Ca^{2+}$ uptake. We first cloned human keratin5 in a mcherry vector and confirmed that transfection of mcherry tagged keratin5 leads to a robust increase in the expression of keratin5 (Fig 5A). Next, we measured resting mitochondrial $Ca^{2+}$ levels and histamine induced mitochondrial $Ca^{2+}$ uptake in empty vector (EV) control and keratin5 overexpression conditions. Interestingly, we observed that although basal mitochondrial $Ca^{2+}$ levels are comparable in 2 conditions, mitochondrial $Ca^{2+}$ uptake was significantly increased upon keratin5 overexpression (Figs 5B, S8A and S8B). This suggests that the increase in keratin5 expression observed upon MCU knockdown could be a compensatory mechanism of the cells to counteract the adverse effects associated with dysregulation of mitochondrial $Ca^{2+}$ homeostasis. We further corroborated our results by examining basal mitochondrial $Ca^{2+}$ levels and mitochondrial $Ca^{2+}$ uptake upon keratin5 silencing. We found that keratin5 knockdown results in a substantial decrease in mitochondrial $Ca^{2+}$ uptake but there is no significant change in the resting mitochondrial $Ca^{2+}$ levels (Figs 5C, S8C and S8D). Next, we examined changes in pigmentation upon keratin5 overexpression and observed that the overexpression of keratin5 resulted in visible and quantitative increase in LD day6 pigmentation (Fig 5D and 5E). Taken together, our data demonstrates that MCU complex driven mitochondrial $Ca^{2+}$ signaling and keratin5 function in a classical feedback loop wherein decrease in mitochondrial $Ca^{2+}$ uptake enhances keratin5 expression while keratin5 in turn augments mitochondrial $Ca^{2+}$ uptake (Fig 5F).

We next investigated if keratin5 works downstream of mitochondrial $Ca^{2+}$ signaling to regulate melanogenesis. We therefore examined if keratin5 can modulate MCUb knockdown induced hyperpigmentation phenotype. We performed either MCUb silencing alone or along with keratin5 knockdown in the LD pigmentation model. As expected, MCUb silencing enhanced pigmentation and keratin5 knockdown in MCUb silenced cells led to rescue of the pigmentation phenotype and corresponding melanin content (Fig 5G and 5H). To understand the molecular mechanism driving the rescue of pigmentation phenotype, we studied

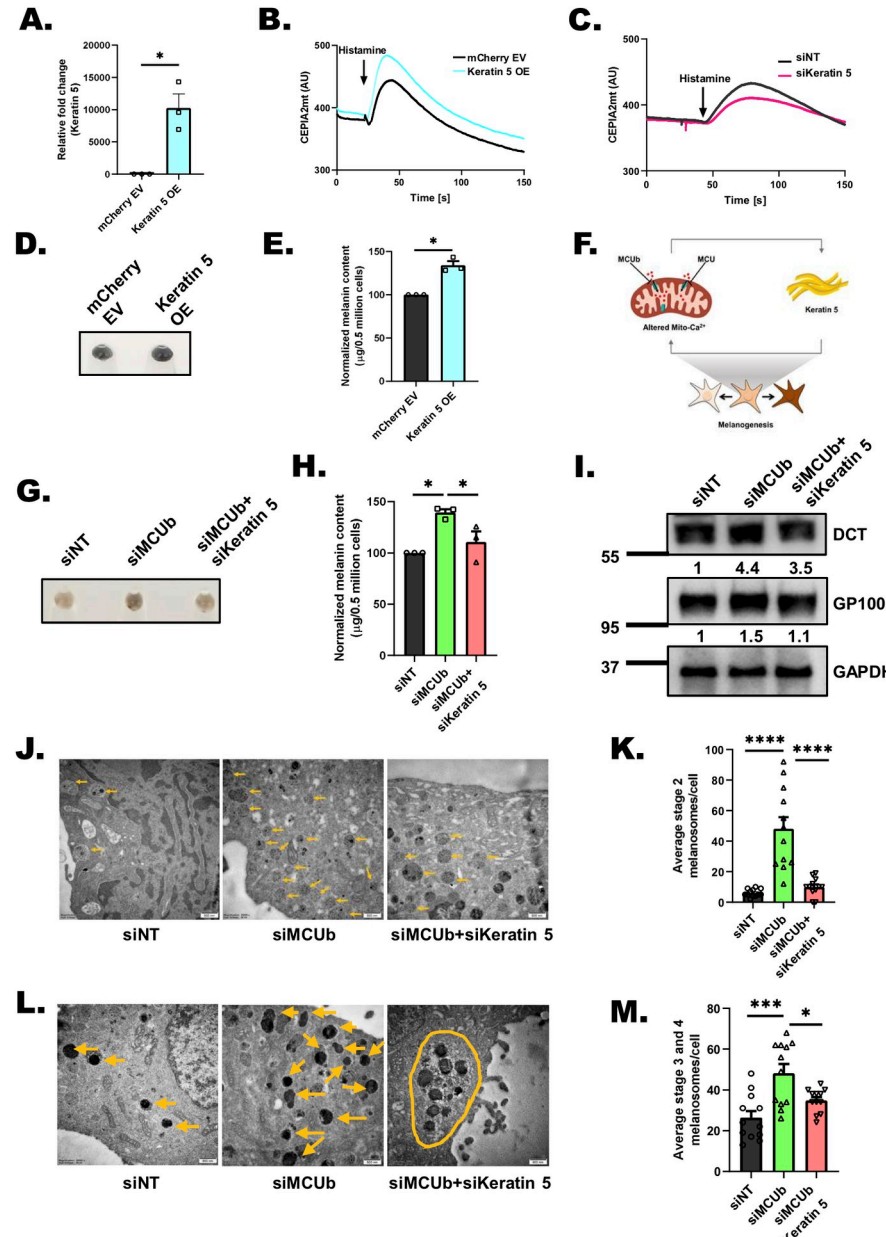

**Fig 5. Keratin5 regulates mitochondrial Ca²⁺ uptake, melanosome maturation, and positioning. (A)** qRT-PCR analysis showing increase in Keratin5 mRNA expression upon ectopic expression of Keratin5 in B16 cells ($N = 3$). **(B)** Representative mitochondrial Ca²⁺ imaging traces of CEPIA2mt in mCherry EV control and Keratin5 overexpressing (OE) cells stimulated with 100 μm histamine. **(C)** Representative mitochondrial Ca²⁺ imaging traces of CEPIA2mt in siNT control and siKeratin 5 cells stimulated with 100 μm histamine. **(D)** Representative pellet images of mCherry EV control and Keratin5 overexpressing (OE) cells on LD day 6 ($N = 3$). **(E)** Melanin content estimation of mCherry EV control and Keratin5 overexpressing (OE) cells on LD day 6 ($N = 3$). **(F)** Mitochondrial Ca²⁺ signaling and keratin5 regulate each other in a feedback loop to ensure optimal melanogenesis. **(G)** Representative pellet images of siNT control, siMCUb, and siMCUb+ siKeratin 5 on LD day 6 ($N = 3$). **(H)** Melanin content estimation of siNT, siMCUb, and siMCUb+ siKeratin5 on LD day 6 ($N = 3$). **(I)** Representative western blot showing expression of DCT and GP100 on LD day 6 upon MCUb, siMCUb+ siKeratin5 silencing as compared to non-targeting control. Densitometric analysis using ImageJ is presented below the blot. **(J)** TEM images of siNT control, siMCUb, and siMCUb+ siKeratin5 B16 LD day 6 cells. Yellow arrows correspond to stage 2 melanosomes in these cells on LD day 6. **(K)** Quantification of number of stage 2 melanosomes/cell in siNT control, siMCUb and siMCUb+ siKeratin5 B16 LD day 6 cells ($N = 12$ cells/condition). **(L)** TEM images of siNT control, siMCUb, and siMCUb+ siKeratin5 B16 LD day 6 cells. Yellow arrows correspond to melanin rich mature (stages 3 and 4) melanosomes in these cells on LD day 6. Yellow oval shape

shows clustering of melanosomes in siMCUb+ siKeratin5. **(M)** Quantification of number of mature melanosomes/cell in siNT control, siMCUb, and siMCUb+ siKeratin5 B16 LD day 6 ($N$ = 12 cells/condition). Data presented are mean ± SEM. For statistical analysis, one sample $t$ test was performed for panel A, E while one-way ANOVA followed by Tukey's post hoc test was performed for panels H, K, M using GraphPad Prism software. Here, * $p < 0.05$; *** $p < 0.001$; and **** $p < 0.0001$. The data underlying for panels A, B, C, E, H, K, and M shown in the figure can be found in S1 Data. EV, empty vector; LD, low density; TEM, transmission electron microscope.

expression of key melanogenic proteins upon MCUb plus keratin5 knockdown and compared it with MCUb silencing alone. As presented in **Fig 5I**, co-silencing of keratin5 plus MCUb inhibits the increase in GP100 and DCT expression observed in siMCUb condition.

Finally, we carried out ultrastructural studies using transmission electron microscope (TEM) for investigating the effect of MCUb knockdown and MCUb plus keratin5 silencing on melanosome biogenesis, maturation, and positioning. We observed that MCUb silencing led to a significant increase in melanosome biogenesis (i.e., number of stage II melanosomes) (**Fig 5J and 5K**) and maturation (i.e., number of stages III and IV melanosomes) (**Fig 5L and 5M**). While simultaneous knockdown of MCUb and keratin5 leads to a substantial decrease in melanosome biogenesis (**Fig 5J and 5K**) and maturation (**Fig 5L and 5M**) as compared to MCUb silencing alone. Further, in case of MCUb silencing alone melanosomes were distributed throughout cells, whereas upon MCUb plus keratin5 knockdown melanosomes were typically clustered together in clumps (**Fig 5L**). It is important to highlight that similar clustering of melanosomes is reported in case of human samples of epidermolysis bullosa simplex with mottled pigmentation [36], a clinical condition associated with keratin5 mutations. Taken together, our data demonstrate that mitochondrial $Ca^{2+}$ signaling regulates keratin5 expression and that in turn modulates melanosome biogenesis, maturation, and positioning (**Fig 5F**).

## MCU drives Keratin 5 expression via NFAT transcription factor

We next investigated the molecular choreography that connects mitochondrial $Ca^{2+}$ signaling to keratin5 expression. Since we observed differential mRNA expression of keratin5 upon silencing of MCU and MCUb, we focused on transcriptional regulation of keratin5. A recent study suggested that downstream of mitochondrial $Ca^{2+}$ signaling NFAT1/4 (Nuclear Factor of Activated T cells) transcription factors get activated [41]. The authors reported that MCU silencing leads to NFAT1/4 activation and their nuclear translocation [41]. Therefore, we bioinformatically scanned the keratin5 promoter for potential NFAT binding sites. We used multiple bioinformatics tools, namely ContraV3 [42], EPD-Search Motif Tool [43], PSCAN [44], and JASPAR 2022 [45], to scan keratin5 promoter for putative binding sites for $Ca^{2+}$ activated NFAT1-4 transcription factors.

We observed that there are no conversed sites for NFAT3 in keratin5 promoter. Although there are couple of potential binding sites for NFAT1 and NFAT4 in keratin5 promoter, they are neither predicted by all 4 bioinformatics tools with confidence nor they are highly conserved. While our survey for NFAT2 binding site (**Fig 6A**) on keratin5 promoter led to identification of 3 highly conserved sites (−946 to −933; −924 to −913; and +83 to +94) and 1 mouse specific site (−547 to −533 from transcription start site) (**Fig 6B**). These sites were predicted by all 4 bioinformatics tools with high confidence. This suggests that MCU can potentially enhance keratin5 transcription through increased activation of NFAT2. Our RNA seq data showed that along with keratin5, keratin7, and 8 are also differentially regulated upon changes in mitochondrial $Ca^{2+}$ signaling. Therefore, we examined keratin7 and 8 promoters for putative NFAT2 binding sites. We found 6 and 2 conserved as well as high confidence NFAT2 binding sites on keratin7 and 8 promoters, respectively (**S9A–S9D Fig**).

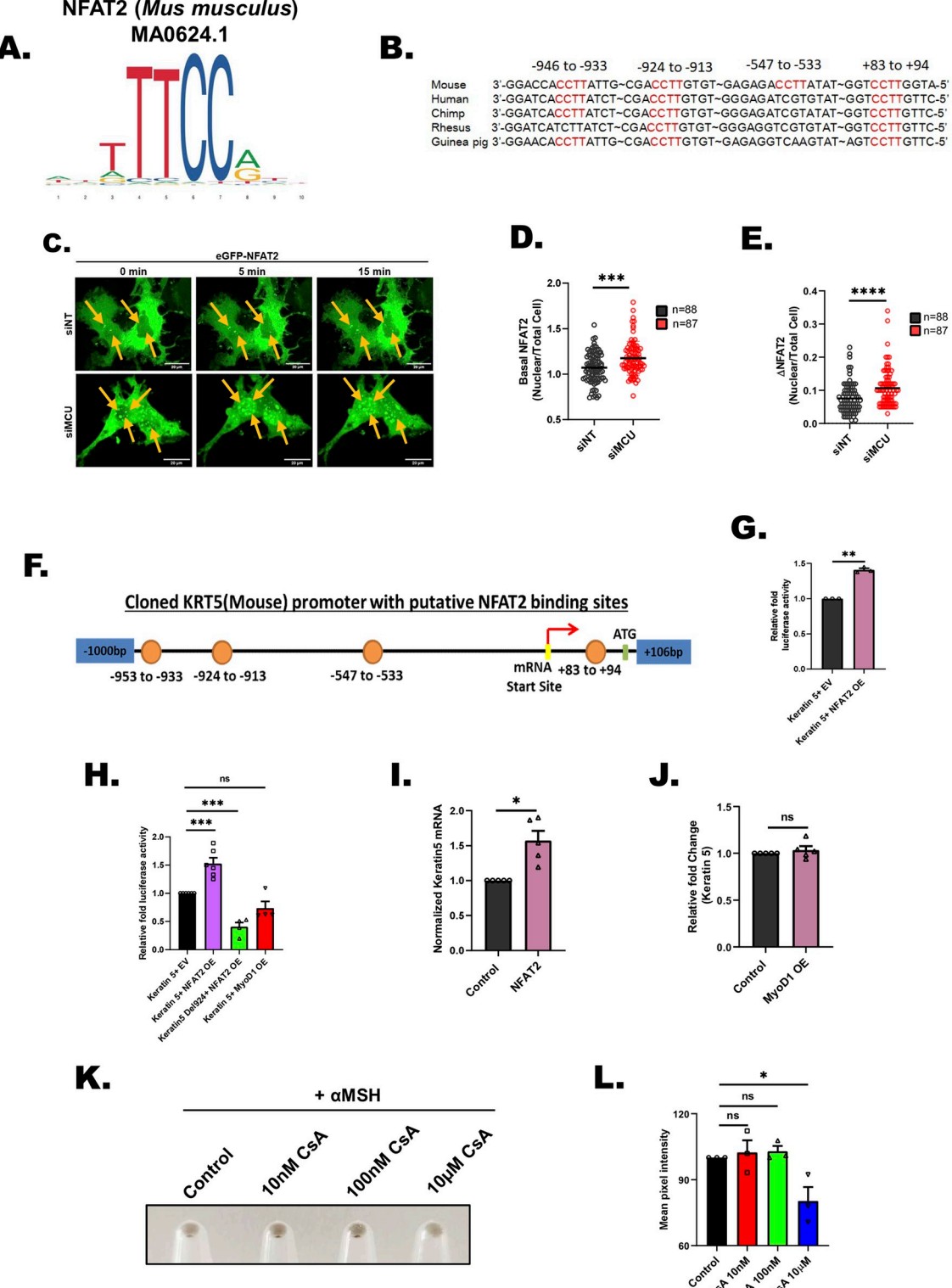

**Fig 6. NFAT2 connects MCU to Keratin5 expression. (A)** Position weight matrix of mouse NFAT2 consensus binding sequence. **(B)** Multispecies sequence alignment of putative NFAT2 binding sites in the mouse keratin5 (KRT5) core promoter. **(C)** Representative confocal microscopy images of differential NFAT2 nuclear translocation in B16 cells transfected with siNT or siMCU, upon stimulation with 100 μm histamine (5 min and 15 min post stimulation). Images have been captured at 63× (oil) magnification in the GFP channel in a Zeiss confocal microscope. Images presented are maximum intensity projections. **(D)** Quantification of basal

NFAT2 nuclear levels in B16 cells with MCU knockdown. Live cell imaging was performed in B16 cells transfected with siMCU or control siRNA along with eGFP-NFAT2 overexpression. Nuclear/Cytoplasmic ratio of NFAT2 was quantified by determining the GFP fluorescence from the nucleus versus total area of a single cell; "*n*" represents the total number of cells imaged per condition. **(E)** Quantification of NFAT2 nuclear translocation upon histamine stimulation. Live cell imaging was performed in B16 cells transfected with siMCU or control siNT along with eGFP-NFAT2 overexpression. Nuclear/Cytoplasmic ratio of NFAT2 was quantified by determining the GFP fluorescence from the nucleus versus total area of a single cell. NFAT2 nuclear translocation in response to histamine (100 µm) stimulation was quantified by calculating the difference between maximal NFAT2 nuclear/total ratio upon histamine stimulation and basal NFAT2 nuclear/total ratio of unstimulated cells; "*n*" represents the total number of cells imaged per condition. **(F)** Schematic representation of putative NFAT2 binding sites in the mouse KRT5 core promoter cloned in pGL4.23 luciferase vector. **(G)** Normalized luciferase activity of the KRT5 promoter luciferase reporter vector with overexpression of eGFP-NFAT2 or empty vector control in B16 cells ($N = 3$). **(H)** Normalized luciferase activity of the KRT5 promoter luciferase reporter vector with overexpression of eGFP-NFAT2 or empty vector control or MyoD1 transcription factor and KRT5 Del924 promoter luciferase reporter vector with overexpression of eGFP-NFAT2 ($N = 4$–6). **(I)** Normalized mRNA expression of mouse KRT5 gene in B16 cells with overexpression of eGFP-NFAT2 or empty vector control ($N = 6$). **(J)** Normalized mRNA expression of mouse KRT5 gene in B16 cells with overexpression of MyoD1 or empty vector control ($N = 5$). **(K)** Representative pellet images upon **cyclosporine A** treatment in B16 cells as compared to vehicle control. **(L)** Mean pixel intensity of **cyclosporine A** treatment in B16 cells as compared to vehicle control ($N = 3$). Data presented are mean ± SEM. For statistical analysis, one sample *t* test was performed for panels G, I, and J, Mann–Whitney test was performed for panels D, E while one-way ANOVA followed by Dunnette's multiple comparisons test was performed for panels H and L using GraphPad Prism software. Here, * $p < 0.05$; ** $p < 0.01$; *** $p < 0.001$; and **** $p < 0.0001$. The data underlying for panels D, E, G, H, I, J, and L shown in the figure can be found in S1 Data. MCU, mitochondrial calcium uniporter.

To examine the role of NFAT2 in connecting mitochondrial Ca²⁺ signaling to keratin5 expression, we first investigated NFAT2 activation upon MCU silencing. As nuclear translocation is a prerequisite for NFAT transcriptional activity, we examined the effects of MCU knockdown on NFAT2 nuclear translocation. We overexpressed GFP tagged NFAT2 in MCU silenced B16 cells and performed live cell imaging to study NFAT2 nuclear translocation (**Fig 6C**). Under resting conditions, we observed an increase in NFAT2 nuclear translocation in MCU knockdown cells as compared to control siRNA transfected cells (**Fig 6C and 6D**).

This data suggests that knockdown of MCU reconfigures the cellular Ca²⁺ homeostasis leading to NFAT2 activation and nuclear translocation. Further, we found that upon stimulation with histamine, there was a significant enhancement in NFAT2 nuclear translocation in MCU knockdown cells in comparison to control cells (**Fig 6C and 6E**). This clearly demonstrates that reduction in MCU expression results in increased NFAT2 nuclear translocation.

To evaluate the role of NFAT2 in regulating keratin 5 transcription, we cloned keratin5 promoter with putative NFAT2 binding sites in a luciferase vector (**Fig 6F**). We performed in vitro luciferase assay with the keratin5 promoter and observed a significant increase in luciferase activity of the keratin5 promoter upon NFAT2 overexpression (**Fig 6G**). This shows that NFAT2 acts as a transcriptional regulator of keratin5. We substantiated NFAT2's role in driving keratin5 promoter activity by generating a deletion mutant for the −924 to −913 NFAT2 binding site and found that the luciferase activity in this mutant is reduced by over 50% (**Fig 6H**). As an additional control, we performed luciferase assays with an unrelated transcription factor (MyoD1) and detected no change in the keratin5 promoter activity. This suggests that the increase in keratin5 promoter activity is NFAT2 specific and it is not driven by any random transcription factor (**Fig 6H**). We further examined mRNA expression of keratin5 upon NFAT2 and MyoD1 overexpression. We observed an evident increase in keratin5 mRNA expression upon ectopic expression of NFAT2 (**Fig 6I**) **but not MyoD1 (Fig 6J)** thereby validating a specific regulation of keratin5 expression by NFAT2. Finally, we examined the consequence of NFAT inhibition by targeting its upstream activator calcineurin with cyclosporine A, an FDA approved calcineurin inhibitor. Our dose-dependent experiments suggest that cyclosporine A treatment at 10 nM and 100 nM concentrations does not alter αMSH stimulated pigmentation while at 10 µm concentration cyclosporine A inhibits αMSH-induced pigmentation (**Fig 6K and 6L**). In support, an earlier study has reported that cyclosporine A

treatment on primary human melanocytes exhibited a decrease in pigmentation and tyrosinase activity [46]. Taken together, our extensive bioinformatics analysis, keratin5 promoter reporter assays, mRNA expression, and NFAT inhibition data clearly demonstrate that NFAT2 acts as a bridging transcription factor that connects MCU silencing to keratin5 expression.

### Mitoxantrone inhibits physiological pigmentation

Next, we investigated the potential of targeting MCU mediated mitochondrial $Ca^{2+}$ uptake in modulating pigmentation. We used mitoxantrone, an FDA approved drug that is highly specific MCU blocker [47], to examine the significance of inhibiting MCU in controlling melanogenesis. We first examined if mitoxantrone can inhibit MCU-mediated mitochondrial $Ca^{2+}$ uptake in B16 cells. We used FURA-2AM and CEPIA2mt to measure αMSH-induced cytosolic $Ca^{2+}$ signals and mitochondrial $Ca^{2+}$ uptake, respectively. We observed that mitoxantrone treatment results in a significant increase in αMSH stimulated cytosolic $Ca^{2+}$ levels (**Fig 7A and 7B**) and a corresponding reduction in mitochondrial $Ca^{2+}$ uptake (**Fig 7C and 7D**), thereby suggesting that mitoxantrone inhibits MCU mediated mitochondrial $Ca^{2+}$ uptake in our model system (**Fig 7A–7D**). Next, we used αMSH for inducing pigmentation in B16 cells and co-treated cells with mitoxantrone to evaluate contribution of MCU in αMSH stimulated pigmentation. Excitingly, we observed that mitoxantrone treatment results in substantial reduction in αMSH stimulated physiological melanogenesis (**Fig 7E**). We further quantitated these phenotypic changes and observed around 70% decrease in αMSH-induced melanogenesis upon mitoxantrone treatment (**Fig 7F**). This data suggests that MCU-mediated mitochondrial $Ca^{2+}$ influx can be targeted for modulating pigmentation.

In summary, we have identified a de novo critical role of mitochondrial $Ca^{2+}$ in driving pigmentation using several independent in vitro and in vivo model systems. Our data demonstrates that mitochondrial $Ca^{2+}$ signaling is a crucial regulator of melanosome biogenesis and maturation. Further, by performing unbiased RNA seq analysis, we have deciphered the molecular mechanism that connects mitochondrial $Ca^{2+}$ signaling to melanogenesis. We reveal that mitochondrial $Ca^{2+}$ signaling regulates transcription of keratin filaments via NFAT2 transcription factor. The keratin filaments in turn contribute to melanosome biogenesis and maturation (**Fig 8**). Importantly, we report that keratin5 augments mitochondrial $Ca^{2+}$ uptake. Therefore, MCU-NFAT2-Keratin5 signaling cascade works as a negative feedback loop that maintains mitochondrial $Ca^{2+}$ homeostasis and ensures optimal pigmentation. Finally, we demonstrate that blocking of MCU results in inhibition of pigmentation. Hence highlighting the potential of targeting MCU for clinical management of pigmentary disorders.

## Discussion

Ionic homeostasis plays an important role in regulating human skin pigmentation [48,49]. Physiologically relevant doses of UV light induce rise in cytosolic $Ca^{2+}$ levels, which leads to enhanced melanin synthesis [50]. Further, our earlier work demonstrated that melanogenic stimuli αMSH increases cytosolic $Ca^{2+}$ levels by inducing ER $Ca^{2+}$ release [25]. This in turn activates STIM1 oligomerization and enhances MITF driven melanogenesis. Interestingly, we recently reported that MITF transcriptionally regulates STIM1 expression in melanocytes and their expression concomitantly increases in sun-exposed tanned human skin [26]. This suggests that a positive feed forward loop driven by ER $Ca^{2+}$ release is a key regulator of melanogenesis. In cellular systems, a major proportion of $Ca^{2+}$ released by ER is taken up by mitochondria via MCU complex [13]. However, the functional relevance of mitochondrial $Ca^{2+}$ uptake in regulating melanogenesis remains totally unappreciated. Here, we report a novel role of mitochondrial $Ca^{2+}$ signaling in pigmentation biology. Using 2 independent cellular

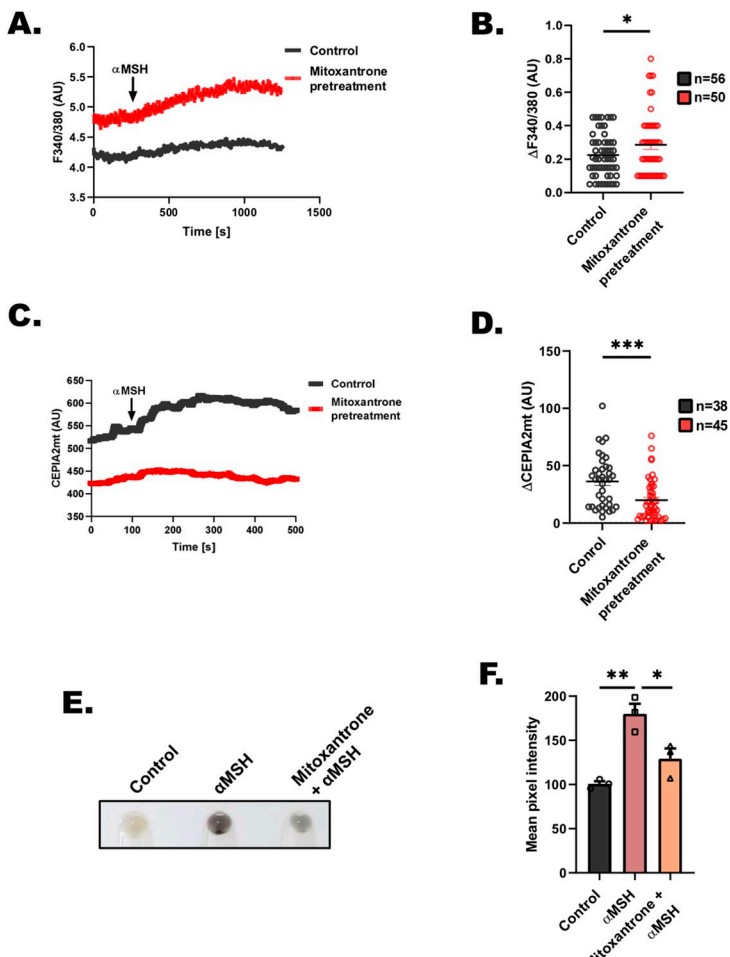

**Fig 7. Mitoxantrone inhibits αMSH-induced physiological pigmentation. (A)** Representative traces of Fura-2 imaging to measure cytosolic $Ca^{2+}$ in control and mitoxantrone pretreated B16 cells stimulated with 1 μm αMSH. **(B)** Quantitation of cytosolic $Ca^{2+}$ levels in control and mitoxantrone pretreated B16 cells stimulated with 1 μm αMSH where "*n*" denotes the number of ROIs. **(C)** Representative mitochondrial $Ca^{2+}$ imaging traces of CEPIA2mt in control and mitoxantrone pretreated B16 cells stimulated with 1 μm αMSH. **(D)** Quantitation of mitochondrial $Ca^{2+}$ uptake by calculating increase in CEPIA2mt signal (ΔCEPIA2mt) in control and mitoxantrone pretreated B16 cells stimulated with 100 μm histamine where "*n*" denotes the number of ROIs. **(E)** Representative pellet pictures upon αMSH, αMSH +mitoxantrone treatment in B16 cells as compared to vehicle control. **(F)** Mean pixel intensity of αMSH, αMSH +mitoxantrone treatment in B16 cells as compared to vehicle control (*N* = 3). Data presented are mean ± SEM. For statistical analysis, unpaired Student's *t* test was performed for panels B and D, while one-way ANOVA followed by Tukey's post hoc test was performed for panel F using GraphPad Prism software. Here, * $p < 0.05$; ** $p < 0.01$; *** $p < 0.001$; and **** $p < 0.0001$. The data underlying for panels A, B, C, D, and F shown in the figure can be found in S1 Data.

models and 2 distinct in vivo systems, we demonstrate a critical role of mitochondrial $Ca^{2+}$ uptake in melanogenesis. Our data from mouse LD pigmentation model and primary human melanocytes suggests that the mitochondrial $Ca^{2+}$ uptake is directly proportional to the extent of pigmentation (**Fig 1**). Importantly, silencing and overexpression of MCU leads to abrogation and augmentation of melanogenesis, respectively (**Fig 2**). Further, knockdown of negatively regulator of mitochondrial $Ca^{2+}$ influx results in increase in melanogenesis (**Fig 2**). Moreover, we validated a critical role of MCU in regulating melanogenesis in vivo by performing MCU silencing, overexpression of human MCU and rescue studies in zebrafish (**Fig 3**). Additionally, MCU$^{+/-}$ mice show reduced, whereas MCUb$^{-/-}$ mice display enhanced epidermal

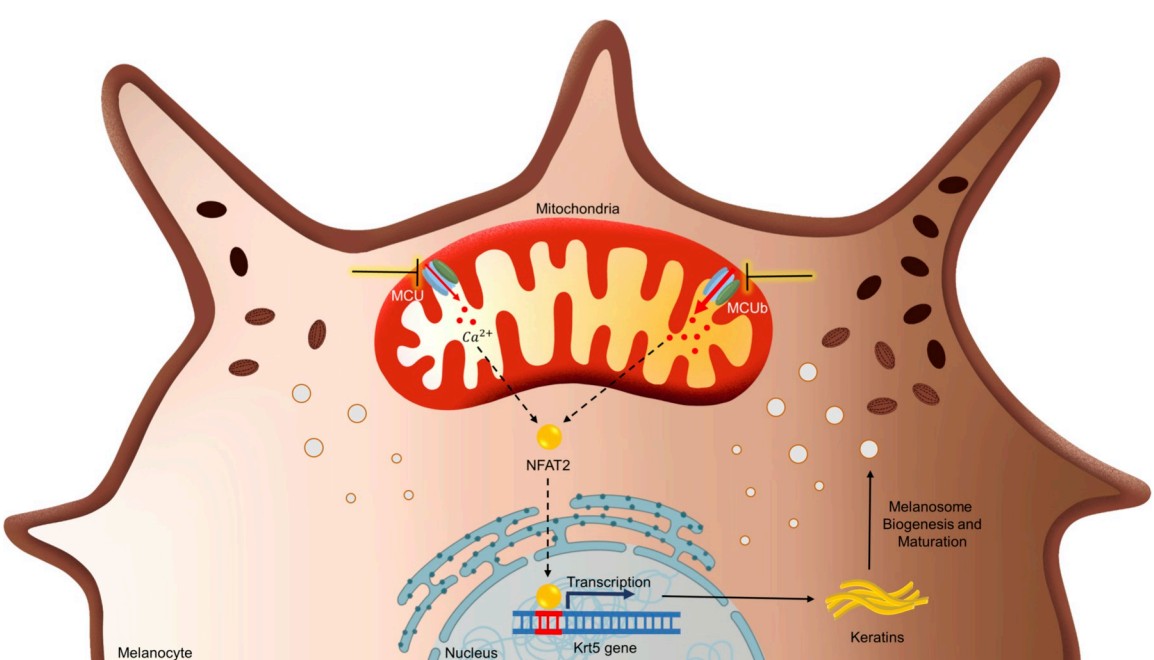

**Fig 8. MCU complex mediated mitochondrial Ca$^{2+}$ dynamics regulates pigmentation via NFAT2-Keratin5 signaling module.**
Graphical summary of the work illustrating that silencing of MCU decreases melanogenesis while MCUb knockdown enhances melanogenesis. Transcription factor NFAT2 gets activated upon MCU silencing and that in turn induces keratin5 expression. Keratin5 drives melanogenesis by augmenting melanosome biogenesis and maturation. MCU, mitochondrial calcium uniporter.

pigmentation (**Fig 3**). Collectively, all this data establishes MCU-mediated mitochondrial Ca$^{2+}$ influx as a novel positive regulator of vertebrate pigmentation.

To characterize the molecular choreography working downstream of mitochondrial Ca$^{2+}$ dynamics for regulating melanogenesis, we performed RNA seq on MCU and MCUb silenced B16 cells (**Fig 4**). Our unbiased RNA seq analysis led to identification of keratin5, 7, and 8 as the bridges that connect mitochondrial Ca$^{2+}$ signaling to melanosome biogenesis and maturation (**Figs 4 and 5**). So far, role of keratin filaments in melanocytes remains largely unappreciated. Keratins provide cytoskeletal scaffold that helps in organelle biogenesis, motility, and function [38,40,51]. Although an earlier study had reported presence of intermediate filaments in melanocytes [52], their role in regulating melanosome biogenesis and/or maturation remains unknown. We here reveal that downstream of mitochondrial Ca$^{2+}$ signaling, keratin5 positively regulates melanosome biogenesis and maturation (**Fig 5J–5M**). To best of our understanding, this is the first report on functional characterization of melanocytic keratins in pigmentation. Taken together, our data demonstrate that keratin5 regulates pigmentation by controlling melanosome biogenesis and maturation.

One of the intriguing findings of our study is that MCU silencing leads to NFAT2 nuclear translocation and consequent increase in keratin5 transcription (**Fig 6**), which in turn augments mitochondrial Ca$^{2+}$ uptake (**Figs 5B, S8A and S8B**). The role of keratins in regulating mitochondrial Ca$^{2+}$ uptake is not reported yet. We here show that keratin5 positively regulate mitochondrial Ca$^{2+}$ uptake. Further, keratin5 enhance melanogenesis by contributing to melanosome biogenesis and maturation. Therefore, this signaling module acts as a classical negative feedback loop that sustains both mitochondrial Ca$^{2+}$ homeostasis and melanogenesis. Although keratin5 levels increase downstream of MCU silencing, the enhanced keratin5 expression is not enough to completely rescue the decrease in pigmentation observed upon

MCU knockdown. It suggests that this signaling module is most likely an outcome of cell compensatory mechanism to maintain mitochondrial $Ca^{2+}$ homeostasis. Indeed, viable MCU global knockouts mice have been reported to acquire compensatory mechanisms to survive in absence of MCU [18,19]. Furthermore, global MCU knockouts have been demonstrated to show cell-specific compensatory outcomes when compared with transient MCU silencing and/or conditional tissue-specific knockouts in adult mice [53–58]. In future, it would be interesting to investigate contribution of the signaling module discovered here in the compensation observed in total MCU knockout mice. Similarly, further studies on the crosstalk between mitochondrial $Ca^{2+}$ signaling and intermediate filaments would provide insights on the role of this signaling cascade in human physiology.

Blockage of NFAT activity demonstrates that cyclosporine A has a dose-dependent effect on αMSH stimulated melanogenesis (**Fig 6K and 6L**). Similar results were earlier reported, wherein cyclosporine A decreased pigmentation in primary human melanocytes [46]. Cyclosporine A is clinically used as an immunosuppressant and literature survey suggest that cyclosporine A intake can induce pigmentation by acting on immune cells. It is important to highlight that immune cell activation is associated with hypopigmentation and indeed, vitiligo (de-pigmentary disorder) is an outcome of enhanced auto-immunity [59,60]. Therefore, systemic administration of cyclosporine A might enhance pigmentation via its action on immune cells. This observation may not be directly associated with cyclosporine A's action on melanocytes. Actually, the discrepancy in the action of cyclosporine A on immune cells and melanocytes has been reported earlier [46]. It was shown that cyclosporine A treatment on primary human melanocytes results in decrease in pigmentation and tyrosinase activity [46]. Taken together, our data, earlier reports on primary melanocytes and clinical studies on patients demonstrate that the effect of cyclosporine A is cell type, drug concentration, and context dependent.

In this study, we have focused on understanding the role of keratin intermediate filaments in regulating pigmentation downstream of mitochondrial $Ca^{2+}$ uptake. However, in our transcriptomics analysis, we observe that several other signaling modules are differentially regulated upon MCU and MCUb silencing (**Figs 4D and S7B**). Therefore, NFAT2-Keratin5 axis can be one of the multiple signaling cascades that work downstream of mitochondrial $Ca^{2+}$ signaling to regulate pigmentation. Going forward, it would be worth to meticulously investigate the contribution of these differentially regulated pathways in vertebrate pigmentation.

Intriguingly, changes in the mitochondrial function including mitochondrial $Ca^{2+}$ dynamics can modulate Store-Operated Calcium Entry (SOCE) mediated by plasma membrane localized Orai channels [41,61,62]. Importantly, SOCE in turn can activate and induce NFAT nuclear translocation [41,63]. Therefore, it could be possible that downstream of changes in the mitochondrial $Ca^{2+}$ uptake, SOCE contributes to NFAT2 activation and nuclear translocation in melanocytes. However, further studies will be required to precisely delineate the role of SOCE in connecting changes in the mitochondrial $Ca^{2+}$ uptake to NFAT2 activation and vertebrate pigmentation.

Another interesting observation of our study is clumping of melanosomes upon keratin5 silencing (**Fig 5L**). Notably, aberrant distribution and clustering of cellular organelles is reported in skin cells of patients with keratin mutations [64]. These observations suggest that mitochondrial $Ca^{2+}$ signaling could be potentially targeted for managing pigmentary disorders especially the ones associated with keratin mutations and/or altered keratin expression. Indeed, we demonstrate that inhibition of MCU with an FDA approved drug (mitoxantrone) leads to a significant decrease in αMSH stimulated physiological melanogenesis (**Fig 7**). Going forward, it would be worth to investigate expression and activity of MCU complex in different types of pigmentary disorders. Moreover, it is important to highlight that a variety of FDA

approved drugs, which can modulate MCU complex activity have been characterized [65,66]. Therefore, future studies aimed at investigating the efficacy of these drugs in regulating pigmentation and thereby alleviating pigmentary disorders as well as aging associated hyperpigmentation are warranted.

Taken together, this study demonstrates a crucial role of mitochondrial $Ca^{2+}$ signaling in pigmentation, identify the molecular mechanisms that connect mitochondrial $Ca^{2+}$ dynamics to melanosome biogenesis/maturation and reveal the potential of targeting mitochondrial $Ca^{2+}$ uptake for managing pigmentation. Furthermore, we report a novel MCU-NFAT2-Keratin5 mediated feedback loop that maintains mitochondrial $Ca^{2+}$ homeostasis and ensures optimal melanogenesis. Given the role of mitochondrial $Ca^{2+}$ signaling and intermediate filaments in cellular physiology, this signaling module could be operational in a variety of cell types. Therefore, it might play an important role in several distinct physiological phenomena and pathological conditions.

## Materials and methods

### Cell culture

B16-F10 cells were obtained from ATCC and cultured in DMEM (Sigma, Bangalore, India). Trypsin, Dulbecco's phosphate buffer saline (DPBS), Versene, fetal bovine serum (FBS), and additional cell culture grade reagents were obtained from Invitrogen, Waltham, Massachusetts, United States of America. Cells were cultured in DMEM medium supplemented with 10% FBS (heat inactivated) at 60% to 80% confluence and at 5% $CO_2$ levels. Lightly and darkly pigmenting neonatal primary human melanocytes (LP and DP, respectively) were procured from Invitrogen, Waltham, Massachusetts, USA. Cells were grown in Medium 254 supplemented with human melanocyte growth supplement-2 and maintained at 37°C in a humidified incubator with 5% $CO_2$ atmosphere. The maintenance and sub culturing of cells were carried out as per the manufacturer's instructions. Cells between passages 3 to 6 were used for experimentation.

### B16 low-density (LD) pigmentation-oscillator model

To set up the LD pigmentation-oscillator model of B16s, cells were seeded at 100 cells/cm$^2$ in DMEM supplemented with 10% FBS as described earlier [24,25].

### siRNA-based transfections

siRNA transfections were performed in T75 flasks on day 3 of the LD pigmentation model, and 100 nM of siRNA (siRNAs from Dharmacon) was added per flask with a 1:3 times V:V ratio of Dharmafect transfection reagent. siRNA and transfection reagent were mixed and incubated over the cells in OptiMEM (Gibco, Waltham, Massachusetts, USA) media for 4 to 6 h for achieving optimal transfection efficiency. Post transfection, the opti-MEM media was removed and then the day 3 culture media was added back to the cells.

siRNA transfections were performed in B16 high density cells using siRNAs from Dharmacon. Dharmafect was used for transfecting B16 cells. siRNA and transfection reagent where mixed and incubated over the cells in OptiMEM (Gibco, Waltham, Massachusetts, USA) media for 4 to 6 h for achieving optimal transfection efficiency. As reported earlier, siRNA transfection in primary melanocytes was done using Nucleofection Kit (Lonza, VPD-1003, U-024 program) [27], and 100 nM of siRNA (smartpool siRNAs from Dharmacon) and 0.5 μg Pmax-gfp plasmid DNA was added per condition. Media was changed after 24 h of transfection. Cells were harvested post 72 h of transfection to capture phenotypic and protein expression changes. The siRNAs were procured from Dharmacon.

The catalog numbers of siRNAs used in the study are included in Table 1.

## qRT-PCR analysis

For mRNA extraction cells were processed with Qiagen RNeasy kit (Catalog #74106). mRNA was then converted to cDNA using high-capacity cDNA reverse transcription kit from Thermo Fisher (Waltham, Massachusetts, USA) (Catalog #4368814). Real-time PCR reactions were performed using SYBR green in Quant Studio 6 Flex from Applied Biosystems. The data were analyzed with Quant Studio real-time PCR software version 1.3. The expression of gene of interest was normalized to that of the housekeeping gene GAPDH. Primers were designed using Primer3 and checked by the NCBI Primer BLAST tool. Gene-specific primers were obtained from Eurofins as given in Table 2.

## Western blotting

Cells were lysed using NP40 lysis buffer supplemented with protease inhibitors. Typically, 50 to 100 μg proteins were subjected to SDS-PAGE (7.5% to 10%). Proteins from gels were then electro-transferred onto PVDF membranes. After blocking with 5% non-fat dry milk (NFDM) dissolved in Tris-buffered saline containing 0.1% Tween 20 (TTBS), blots were probed over-night at 4°C, with specific primary antibodies in TTBS containing 2% NFDM. The primary antibodies used were typically procured from Abcam, Cell signaling and were used at 1:500 to 1:2,000 dilutions. The following day, membranes were incubated for 2 h at room temperature with a horseradish-peroxidase-conjugated anti-mouse or anti-rabbit IgG antibody in TTBS containing 2% NFDM. Detection was performed using the enhanced chemiluminescence reagent (ECL Western blotting detection reagents; Amersham Biosciences). Quantification of bands was performed by densitometry using the ImageJ software. The catalog number and company name for the antibodies are provided in Table 3.

## Tyrosinase activity/DOPA assay

Tyrosinase-enzyme activity was checked in cell lysates by performing DOPA assay as previously reported [67]. Briefly, cell lysates were prepared in NP-40 lysis buffer and an equal amount of protein was run on a gel under nonreducing/native conditions. The gel was then immersed in phosphate buffer supplemented with tyrosinase substrate L DOPA (Sigma Chemicals, Banga-lore, India). Enzyme activity corresponded to the formation of black color pigment.

## Melanin content assay

Melanin content assay was performed as described earlier [68]. The equal number of cells were lysed in 1N NaOH by heating at 80°C for 2 h and then absorbance was measured at 405 nm.

**Table 1. Details of siRNAs used.**

| siRNA | Catalog Number |
|---|---|
| siNT | D-001810-10-20 |
| siMCU (Mouse) | J-062849-05-0010 |
| siMCU (Human) | L-015519-02-0010 |
| siMCUb (Mouse) | L-061742-01-0005 |
| siMCUb (Human) | L-016108-02-0010 |
| siKeratin 7 (Mouse) | L-063818-01-005 |
| siKeratin 8 (Mouse) | L-043866-01-005 |
| siKeratin 5 (Mouse) | L-048162-01-0005 |
| siKeratin 5 (Human) | L-011067-00-0005 |

**Table 2. List of qRT-PCR primers (F stands for forward primer and R for reverse primer).**

| Gene name | Sequence | Species |
| --- | --- | --- |
| MCU F | GGTCGCCATCTACTCACCAG | Mouse |
| MCU R | CAGGGTCTTCACGTCGTTCA | Mouse |
| MCUb F | AGAAGAGCCAAGTGGAGAGC | Mouse |
| MCUb R | AGACCACTGGTGTTGGCTTC | Mouse |
| MICU1 F | TTCTGGCAGAAAATTGCCCAG | Mouse |
| MICU1 R | GGAGAGGACTGTTGTGAGGAA | Mouse |
| MICU2 F | GTCCTTTTGCCATTTTACAACGC | Mouse |
| MICU2 R | TCTCTTGAACTCTGCCAGCC | Mouse |
| EMRE F | GTCAGTCATCGTCACTCGCA | Mouse |
| EMRE R | CCCTGTGCCCTGTTAATCGT | Mouse |
| GAPDH F | AACTGCTTAGCACCCCTGGC | Mouse |
| GAPDH R | ATGACCTTGCCCACAGCCTT | Mouse |
| Keratin 5 F | GGGGGTCTTGGTTTGGGTAG | Mouse |
| Keratin 5 R | TGCAGCAGGCTCTTCTTAGC | Mouse |
| Keratin 7 F | ATCCACTTCAGCTCTCGCTC | Mouse |
| Keratin 7 R | ATCGATGTCCACACTCAGCG | Mouse |
| Keratin 8 F | GATCCAACACTTTCAGCCGC | Mouse |
| Keratin 8 R | GGTTGGCCAGAGGATTAGGG | Mouse |
| NFAT2 F | CACCGCATCACAGGGAAGAC | Mouse |
| NFAT2 R | GCACAGTCAATGACGGCTC | Mouse |
| MCU F | TGGCAGAATTTGGGAGCTGT | Human |
| MCU R | CCCCGATCCTCTTCTTGCAG | Human |
| MCUb F | TGTCAGCTTGTCCCACGTTT | Human |
| MCUb R | GCCCAGGAGTTTGATTTGGC | Human |
| MICU1 F | GCACCACCCATTCAAGCATC | Human |
| MICU1 R | CAGCAGGAGGGAAGTCAAGG | Human |
| MICU2 F | GCGGGATGGCAGTTTTACAG | Human |
| MICU2 R | GCGCTGCTTACGAAGAGAC | Human |
| EMRE F | GCTAGTATTGGCACCCGTCA | Human |
| EMRE R | GAGAACACACGGAGAAGGCC | Human |
| Keratin 5 F | AATGCAGACTCAGTGGAGAAGG | Human |
| Keratin 5 R | TCCAGAGGAAACACTGCTTGTG | Human |
| DCT F | GCCTGGGTGCAGAGTCG | Human |
| DCT R | TAGCCGGCAAAGTTTCCTGT | Human |
| TYR F | CCCATTGGACATAACCGGGA | Human |
| TYR R | TCTTGAAAAGAGTCTGGGTCTGAA | Human |
| GP100 F | GGCTGTGGGGGCTACAAAA | Human |
| GP100 R | GAGGGACACTTGACCACCTCT | Human |
| TYRP1 F | CCGAAACACAGTGGAAGGTT | Human |
| TYRP1 R | TCTGTGAAGGTGTGCAGGAG | Human |
| MyoD1 FP | GCTACGACACCGCCTACTAC | Mouse |
| MyoD1 RP | GAGATGCGCTCCACTATGCT | Mouse |

Equal number of zebrafish embryos were lysed in 1N NaOH by heating at 90°C for 6 h. Melanin content was estimated by interpolating the sample readings on the melanin standard curve (μg/ml) obtained with synthetic melanin. Finally, melanin content is normalized to the respective control sample (siNT or empty vector) and is plotted as a single data point on the bar graphs.

**Table 3. Details of antibodies used.**

| Antibody | Company | Catalog Number |
|---|---|---|
| DCT | Abcam | ab74073 |
| GP100 | Abcam | ab137078 |
| β-Tubulin | Abcam | ab21058 |
| MCU | Cell signaling | D2Z3B-14997 |
| Calnexin | Abcam | ab22595 |
| β-Actin | Cell signaling | CST4967S |

## Mitochondrial calcium imaging

For performing mitochondrial $Ca^{2+}$ imaging, B16 cells ($0.5 \times 10^6$ cells/well) were plated on confocal dishes (SPL Life Sciences, Korea). After 24 h post B16 cell plating (at 60% confluency), pCMV CEPIA2mt (Addgene plasmid # 58218) plasmid (1.5 μg) was overexpressed using Lipofectamine 2000 (Invitrogen, 11668–019). After 24 h post transfection, cells were washed 3 times and bathed in HEPES-buffered saline solution (140 mM NaCl, 1.13 mM MgCl2, 4.7 mM KCl, 2 mM CaCl2, 10 mM D-glucose, and 10 mM HEPES; pH 7.4) for 5 min before $Ca^{2+}$ measurements. Nikon Eclipse Ti2 microscope was used to record and analyze fluorescence images of several cells using 60× oil objective. For CEPIA2mt following excitation/emission wavelengths: (488 nm/500 to 550 nm) were used, and 100 μm histamine in $Ca^{2+}$ free bath solution was used to release $Ca^{2+}$ from the ER via $IP_3Rs$; 1 μm αMSH in $Ca^{2+}$ free bath solution was used to release $Ca^{2+}$ from the ER after serum starving cells. Experiments were performed at least 3 times and the final data are plotted where the number of cells is denoted as "n." For measuring mitochondrial $Ca^{2+}$ on different days in LD pigmentation-oscillator model of B16s, cells were seeded at 100 cells/cm$^2$ on confocal dishes. pCMV CEPIA2mt plasmid (1.5 μg) was overexpressed using Lipofectamine 2000, 24 h before measuring mitochondrial $Ca^{2+}$. pCMV CEPIA2mt was a gift from Masamitsu lino (Addgene plasmid # 58218).

For measuring mitochondrial $Ca^{2+}$ in LP and DP primary human melanocytes using Rhod-2 AM (543 nm/580 to 650 nm) dye, primary human melanocytes ($1 \times 10^6$ cells/dish) were plated on confocal dishes. After 24 h post cell plating (at 50% to 60% confluency), cells were washed with media without FBS and antibiotic-antimycotic agents. Further, cells were incubated in media containing 1 μm Rhod-2 AM at 37°C for 30 min. For measuring mitochondrial $Ca^{2+}$ in LP and DP primary human melanocytes using Rhod-2 AM dye and MitoTracker Green FM dye, primary human melanocytes ($1 \times 10^6$ cells/dish) were plated on confocal dishes. Cells were incubated in media containing 1 μm Rhod-2 AM at 37°C for 30 min. Cells were washed with PBS and incubated in media containing 50 nM MitoTracker Green FM at 37°C for 30 min. Cells were washed and kept in HBSS (HEPES-buffered saline solution (140 mM NaCl, 1.13 mM $MgCl_2$, 4.7 mM KCl, 2 mM $CaCl_2$, 10 mM D-glucose, and 10 mM HEPES, adjusted to pH 7.4 with NaOH)) containing 5 mM $CaCl_2$ for imaging. The cells were stimulated with 100 μm histamine in HBSS containing 5 mM $CaCl_2$. Nikon Eclipse Ti2 microscope was used to record and analyze fluorescence images of several cells using 60× oil objective. The data was acquired every 1 s intervals for 5 to 10 min.

For measuring mitochondrial $Ca^{2+}$ post mitoxantrone treatments, cells were pretreated for 1 h with 1 μm Mitoxantrone (Sigma-Aldrich, M6545) and 1 μm αMSH was used to release $Ca^{2+}$ from the ER. Please refer to **S10 and S11 Figs** for time lapse images of mitochondrial $Ca^{2+}$ uptake in response to Histamine and αMSH (using CEPIA) in B16 cells and mitochondrial $Ca^{2+}$ uptake in response to Histamine (using Rhod-2 AM) in primary human melanocytes (scale = 50 μm).

## Cytosolic calcium imaging

For measuring cytosolic $Ca^{2+}$ on different days in B16 LD pigmentation model, cells were seeded at 100 cells/$cm^2$ on confocal dishes. The cells were incubated with 4 μm ratiometric Fura-2-AM dye (Thermo Fisher Scientific, F1221) for 40 min at 37˚C, washed 3 times, and bathed in HEPES-buffered saline solution (140 mM NaCl, 1.13 mM MgCl2, 4.7 mM KCl, 2 mM CaCl2, 10 mM D-glucose, and 10 mM HEPES; pH 7.4) for 5 min before $Ca^{2+}$ measurements to eliminate excess Fura-2. Nikon Eclipse Ti2 microscope was used to record and analyze fluorescence images of several cells using 20× objective. For Fura-2-AM following excitation/emission wavelengths: (340 and 380 nm alternatively/510 nm) were used. The ratio of fluorescence at 340 and 380 nm ($F_{340}/F_{380}$) was analyzed in cells using NIS-Elements AR 5.30.00 software, and 100 μm histamine in $Ca^{2+}$ free bath solution was used to release $Ca^{2+}$ from the ER via $IP_3Rs$. Experiments were performed at least 3 times and the final data are plotted where the number of cells is denoted as "n".

For measuring cytosolic $Ca^{2+}$ in LP and DP primary human melanocytes, primary human melanocytes ($1 \times 10^6$ cells/dish) were plated on confocal dishes. The cells were incubated with 4 μm ratiometric Fura-2-AM dye (Thermo Fisher Scientific, F1221) for 45 min at 37˚C, washed 3 times, and bathed in HEPES-buffered saline solution for 5 min before $Ca^{2+}$ measurements to eliminate excess Fura-2, and 100 μm histamine in $Ca^{2+}$ free bath solution was used to release $Ca^{2+}$ from the ER via $IP_3Rs$. Experiments were performed at least 3 times and the final data are plotted where the number of cells is denoted as "n".

For measuring cytosolic $Ca^{2+}$ post siMCU and siMCUb transfections, cells were seeded at ($0.2 \times 10^6$ cells/well) on confocal dishes. For measuring mitochondrial $Ca^{2+}$ in MCU overexpressing B16 cells, cells were seeded 24 h before transfection at a density of $0.8 \times 10^6$ cells/well on confocal dishes. Human pDEST47-MCU-GFP plasmid and Cepia2mt plasmid (1.5 μg) were overexpressed in B16 cells plated at 60% confluency using Lipofectamine 2000 (Invitrogen, 11668–019). For measuring cytosolic $Ca^{2+}$ post mitoxantrone treatments, cells were pretreated (for 1 h) with 1 μm Mitoxantrone (Sigma-Aldrich, M6545). Cells were washed 3 times and bathed in HEPES-buffered saline solution for 5 min before $Ca^{2+}$ measurements. Nikon Eclipse Ti2 microscope was used to record and analyze fluorescence images of several cells using 60× oil objective. Experiments were performed at least 3 times and the final data are plotted where the number of cells is denoted as "n".

## MCU overexpression in B16 cells

B16 cells were seeded 24 h before transfection at a density of $1.0 \times 106$ cells/well in 6-well plates. Human pDEST47-MCU-GFP plasmid (1.5 μg) was overexpressed in B16 cells plated at 60% confluency using Lipofectamine 2000 (Invitrogen, 11668–019). After 18 h post transfection, 1 μm α-MSH treatment (Sigma-Aldrich, M4135) was given for 30 h followed by cell termination. Human MCU-GFP plasmid was a gift from Vamsi Mootha (Addgene plasmid # 31732).

## OCR (oxygen consumption rate) measurements

To measure the effect of MCU and MCUb human isoforms overexpression on OCR in B16 cells, cells were cultured overnight on XF24 microplate (Agilent, #100777–004). The day of the experiment, the cells were incubated for 2 h at 37˚C in a non-$CO_2$ incubator in DMEM, supplemented with 1 mM NaPyr, 5 mM glucose, 33 mM NaCl, 15 mg phenol red, 25 mM HEPES, and 1 mM of L-Glu. The OCR was assessed in real-time with the XF24 Extracellular Flux Analyzer (Agilent). For the acquisition, basal respiration was recorded before injecting 1 μm oligomycin (Sigma Aldrich, #75351), followed by 1 μm FCCP, and 1 μm rotenone (Sigma Aldrich,

#45656) plus 1 μm antimycin A (Sigma Aldrich, #A8674). The results were normalized for the fluorescence of Calcein (Sigma-Aldrich, #80011–3). In detail, cells were loaded with 2 μm Calcein for 30 min at 37°C before measuring fluorescence using a Perkin Elmer EnVision plate reader in well-scan mode using a 480/20 nm filter for excitation and a 535/20 nm filter for emission.

## Mitochondrial membrane potential (ΔΨ)

Mitochondrial membrane potential (ΔΨ) was measured with TMRE (Tetramethylrhodamine, Ethyl Ester, Perchlorate; Thermo Fisher, T669). B16 cells were loaded with 200 nM TMRE (excitation/emission: 549/574 nm) for 20 min at 37°C and 5% $CO_2$ levels followed by FACS analysis (BD FACSVerse), and 50 μm FCCP for 10 min (Carbonyl cyanide 4-(trifluoro-methoxy) phenylhydrazone; Sigma Aldrich, C2920) was added as a control. Data was analyzed with the FlowJo software 10.

## ATP measurements

The ENLITEN ATP Assay System (Promega, FF2000) was used as per manufacturer's protocol. Briefly, equal number of B16 cells were harvested and resuspended in PBS. Further cell suspension was mixed with 2.5% trichloroacetic acid (TCA). Tris-acetate buffer (pH 7.75) was then added to neutralize the TCA and to dilute the TCA to a final concentration of 0.1%. The diluted sample (100 μl) was added to an equal volume of rL/L reagent and luminescence was measured (Promega, GLOMAX LUMINOMETER). The ATP standard was serially diluted to generate a regression curve for calculating ATP concentrations in individual samples. The relative ATP concentration was determined and normalized to the control cells.

## Mitoxantrone treatment in αMSH-induced pigmentation assay

In B16 cells, seeded at HD, pigmentation was induced by adding 1 μm αMSH (Sigma-Aldrich, M4135) for 48 h. Cells were pretreated (for 1 h) with 1 μm Mitoxantrone (Sigma-Aldrich, M6545) followed by addition of 1 μm αMSH for 48 h. After 48 h, cells were terminated and pellets were made. Further, mean pixel intensity of pellets was calculated using ImageJ software.

## Generation of KRT5 overexpression construct

Human KRT5 overexpression plasmid was generated by sub cloning human KRT5 cDNA into mCherryC1 vector at EcoRI/XhoI sites. KRT5 cDNA was PCR amplified using pBabe mRFP1-KRT5 (Addgene#59493) as template using Phusion High Fidelity Polyemrase (F503, Thermo). The amplified PCR product was restriction digested using EcoRI/XhoI (NEB). Simultaneously, mCherryC1 empty vector was also digested along with shrimp alkaline phosphatase (NEB) treatment. This was followed by gel purification of digested vector and PCR product. Vector and insert ligation was performed using Rapid DNA ligation Kit (K1422, Thermo). The ligation mix was transformed into chemically competent E. coli DH5α cells followed by plating onto Kanamycin containing agar plates. Positive clones were screened by restriction digestion analysis of plasmid DNA obtained by miniprep. Primers utilized for cloning are as follows:

hKRT5CDS FP 5′-AAACTCGAGCCATGTCTCGCCAGTCAA-3′
hKRT5CDS RP 5′-CCCGAATTCTTAGCTCTTGAAGCTCTTCC-3′.

## Keratin5 overexpression in B16 cells

Human KRT5 overexpression plasmid (1.5 µg) was overexpressed in B16 cells on day 3 of LD-pigmentation model. The effect of KRT5 overexpression was analyzed on LD day 6 by performing melanin-content assays.

## Cloning of Keartin5 promoter and its mutant

Mouse Keratin5 promoter sequence was obtained from Eukaryotic Promoter Database (EPD), which was followed by NCBI-BLAST analysis of the sequence to identify mRNA start site and first codon. Primers were designed to amplify a 1,106 bp region (−1,000 to + 106, w.r.t. to start codon) of the KRT5 promoter. KRT5 core promoter −1,000 to +106 (KRT5PWT) was amplified from B16F10 genomic DNA, isolated using DNeasy Blood and Tissue Kit (69504, Qiagen) as per manufacturer's protocol. This was followed by PCR amplification of the target region using Phusion High Fidelity Polymerase (F503, Thermo), which was further cloned into pGL4.23 luciferase reporter vector (Promega) at the KpnI/HindIII sites. Cloned promoter was verified by restriction digestion and sequencing to verify its identity. Primers utilized for cloning are as follows:

KRT5P FP 5′-TTAAGGTACCGTGTTTGCGGGCGG-3′
KRT5P RP 5′-GGTTAAGCTTTGGCGAGACATGATGG-3′.

Further, deletion mutant of the Keratin5 promoter (KRT5P Del924) was generated at the NFAT2 consensus binding site (-924 to -913) and was verified by sequencing. The following primer set was used for the same:

Del -924 FP 5′-TGGAATAACTGAGAACTGCTTGGGGGGCACCGGG-3′
Del -924 RP 5′-CCCGGTGCCCCCCAAGCAGTTCTCAGTTATTCCA-3′.

## In vitro luciferase assay

B16 cells were seeded 24 h before transfection at a density of $0.5 \times 10^5$ cells/well in 24-well plates. Cells were transfected with KRT5PWT or KRT5P Del924 along with eGFPc1-huNFAT-c1EE-WT (Addgene#24219) as indicated, using Turbofect transfection reagent (R0532, Thermo) as per manufacturer's protocol. Transcription factor MyoD1 (Myod1 (NM_010866) Mouse Tagged ORF Clone (Origene#MR225026)) was transfected in cells along with KRT5PWT. Renilla luciferase control plasmid was utilized for transfection normalization in all experiments; 48 h post transfection, cells were assayed for luciferase activity using the dual luciferase assay kit (E1910, Promega, Madison, Wisconsin) as per manufacturer's protocol. Data is representative of at least 3 biological replicates with 3 technical replicates each.

## Cyclosporine A treatment in αMSH-induced pigmentation

B16 cells were seeded at a density of $0.5 \times 10^5$ cells/well in 6-well plates; pigmentation was induced by adding 1 µm αMSH (Sigma Aldrich, M4135) for 24 h. Cells were then treated with 10 nM, 100 nM, and 10 µm Cyclosporine A (Merck, C1832-5MG) for 24 h. After 24 h, cells were terminated and pellets were made. The mean pixel intensity of pellets was calculated using ImageJ software.

## Mitochondrial content estimation

For mitochondrial content estimation in LD model system, B16 cells were seeded at 100 cells/$cm^2$ on confocal dishes. Cells were then stained with 0.1 µm/100 nM MitoTracker Red FM (Invitrogen, Waltham, Massachusetts, USA) for 30 min at 37°C in complete culture medium, protected from light. This was followed by gentle washing with 1× HBSS 3 times and imaging

**Table 4. Details of morpholino used.**

| Gene name | Sequence |
|---|---|
| Scrambled morpholino | CCTCTTACCTCAGTTACAATTTATA |
| MCU | ATCTACACACTTTCGCAGCCATCTC |

on a Zeiss LSM 880 confocal microscope using 63× oil objective. The same staining protocol was followed for LP and DP primary human melanocytes which were seeded at a density of $2.5 \times 10^4$ cells/confocal dish. Images were analyzed for the fluorescence intensity using FIJI software. Data quantitated from individual cells of multiple biological and technical replicates has been plotted.

## Transmission electron microscopy

Transmission electron microscopy was performed on siNT, siMCUb, and siMCUb+Keratin 5 B16 cells on LD day 6 using standard protocols. Briefly, cells were fixed overnight in fixative containing 2.5% glutaraldehyde and 4% paraformaldehyde, gradually dehydrated in graded series of ethanol and embedded in Epon 812 resin. Ultrathin sections were cut and stained with uranyl acetate and lead citrate and images were captured using a transmission electron microscope (JEM-1400Flash).

## Zebrafish husbandry

Zebrafish used in this study were housed in CSIR-Institute of Genomics and Integrative Biology, India with proper ethical protocols approved by the Institutional Animal Ethics Committee of CSIR-Institute of Genomics and Integrative Biology, India (Protocol no. GENCODE-C-24) with care to minimize animal suffering.

## Morpholino design and microinjections

Antisense morpholino (MO) oligonucleotides were designed against MCU of zebrafish to block the translation and were ordered from Gene Tools, USA. The MO were reconstituted in nuclease free water (Ambion, USA) according to the recommended protocol by GENE tools to get 1 mM concentration and was stored in −20°C for further experiments. MO was injected at a single-cell stage and embryos were cultured in E3 water at 28°C. Dose titration for MO was done using 3 different concentrations (100, 200, 500 μm) and the optimal dose concentration 100 μm was selected based on survival and phenotype percentage. Scrambled MO was used as a negative control for the experiments. The embryos were screened for pigmentation phenotype at 30 hpf, 36 hpf, and 48 hpf. Imaging was done using Nikon microscope. The sequence of morpholino used is provided in Table 4.

## Overexpression and complementation/rescue assay of MCU in zebrafish

Plasmid containing CDS sequences of human MCU were taken from Addgene and restriction digestion was performed. In vitro transcription (IVT) was performed using T7 mMassage mMachine kit (Invitrogen) and purified using RNA purification column (Roche). MCU-IVT was injected at single-cell stage and embryos were cultured in E3 water. Dose titration with different concentration of IVT (50, 100, 200 ng/μl) was performed for the overexpression experiment. The optimal concentration 50 ng/μl was selected based on survival and phenotype percentage of the embryos. The embryos with overexpression were screened for pigmentation phenotype at 24 hpf and 48 hpf.

Complementation/rescue assays were performed in zebrafish using MCU knockdown embryos. A cocktail of MCU MO (100 μm) and Human MCU IVT (50 ng/μl) was injected into single-cell stage zebrafish. RFP IVT was used as a negative control for the complementation assay experiment. The embryos were screened for pigmentation phenotype at 24 hpf and 48 hpf to understand the level of rescue in complementation assay. Imaging was done using Nikon microscope.

## MCU$^{+/-}$ and MCUb$^{-/-}$ mice

The heterozygous MCU knockout (MCU$^{+/-}$) mice and complete MCUb knockout (MCUb$^{-/-}$) were generated and reported in our earlier studies [31,34]. In order to phenotypically assess the role of MCU and MCUb in mice epidermal pigmentation, we took tail images of MCU$^{+/-}$, MCUb$^{-/-}$, and corresponding age matched wild-type control mice with a digital camera. To quantitate the differences in the epidermal pigmentation between WT mice and transgenic (MCU$^{+/-}$ and MCUb$^{-/-}$) mice, we calculated mean pixel intensity of mice tail using ImageJ software. We analyzed multiple sites from 7 independent tails of wild-type mice, MCU$^{+/-}$ and MCUb$^{-/-}$ mice. We averaged the mean pixel intensity from each tail and used it as a single data point for statistical comparison.

## Mitochondrial calcium measurements in skin fibroblasts isolated from MCU$^{+/-}$, MCUb$^{-/-}$ and corresponding wild-type control mice

Mitochondrial Ca$^{2+}$ was measured in adult primary skin fibroblasts from wild-type, MCU$^{+/-}$, and MCUb$^{-/-}$ mice plated on 24 mm glass coverslips, via transduction of the fluorescent probe 4mtGCaMP6f with adenovirus infection; 48 h later, coverslips were transferred to an imaging chamber and incubated in 1 ml of modified KRB buffer (supplemented with 1 mM CaCl2). Imaging was performed on an inverted microscope (Zeiss Axiovert 200), equipped with a 40× /1.3 N.A. PlanFluor objective. Excitation was performed with a Deltaram V high-speed mono-chromator (Photon Technology International) equipped with a 75W Xenon Arc lamp. Images were captured using an Evolve 512 Delta EMCCD (Photometrics), and the system was controlled by MetaMorph 7.5 (Molecular Devices), assembled by Crisel Instruments. The GCaMP6f family is a collection of GFP-based green fluorescent indicators enabling the detection of fast Ca$^{2+}$ transients. To measure resting mitochondrial Ca$^{2+}$, a GCaMP probe targeted to the mitochondrial matrix was generated (4mtGCaMP6f) [69]. This is a ratio-metric probe and cells were alternatively illuminated every second at 490 and 410 nm, and fluorescence was collected through a 525/50 filter (Chroma) mounted on an OptoSpin25 wheel (Cairn research). The exposure time was 100 msec, and 100 μm histamine was added to examine maximal mitochondrial Ca$^{2+}$ uptake. Analysis was performed with the Fiji distribution of ImageJ. Data are presented as a fluorescent ratio (490/410 nm) after frame-by-frame background correction.

## RNA sequencing

LD day 6 samples siNT, siMCU, and siMCUb were sent for mRNA sequencing to Clevergene, Bengaluru, India. The sequence data was generated using Illumina NovaSeq. Data quality was checked using FastQC and MultiQC software. GRCm39 was used as the reference genome. Gene level expression values were obtained as read counts using feature-counts software and normalized counts were obtained. Fold change was calculated for siMCU and siMCUb samples with respect to non-targeting siRNA control. Genes showing log2 fold change more than +1 in expression with respect to siNT control were taken as up-regulated genes in siMCU and siMCUb. While a cut off of log2 fold change less than −1 was set for determining down-

regulated genes in siMCU and siMCUb. These up-regulated and down-regulated genes were analyzed for pathway enrichment using DAVID software. Pathway enrichment analysis was performed based on the GO database. Enriched pathways were plotted with enrichment scores in Graphpad. To determine oppositely regulated pathways in siMCU and siMCUb, common pathways up in siMCU and down in siMCUb and vice versa (down in siMCU and up in siM-CUb) were plotted. Further, to get common but oppositely regulated genes, the Venn diagram was plotted for genes up in siMCU and down in siMCUb and vice versa.

### Measurement of NFAT2 nuclear translocation

To evaluate the effects of MCU mediated $Ca^{2+}$ flux on NFAT2 nuclear translocation, B16 cells were seeded at a density of 20,000 cells/ well in a 6-well plate. Cells were then transfected with smart pool siRNA against mouse MCU (Dharmacon) or control siRNA (Dharmacon) at a concentration of 100 nM using DharmaFect transfection reagent according to manufacturer's protocol; 24 h post transfection cells were reseeded into glass bottom imaging dishes, and 72 h post transfection cells from both siMCU and siNT conditions were transfected with 1 μg of eGFP-NFAT2 overexpression plasmid (Addgene#24219) using Lipofectamine 2000 reagent according to manufacturer's protocol, and 96 h post siRNA transfections live cell imaging was performed. Cells were washed thrice with $Ca^{2+}$ containing HBSS and imaged in $Ca^{2+}$ containing HBSS bath using a Nikon Eclipse Ti2 microscope at 40× magnification. eGFP-NFAT2 was excited using 488 nm laser and emission was captured at 510 nm. eGFP-NFAT2 translocation in response to 100 μm histamine stimulation was analyzed in real-time using the equation:

$$NFAT2\ translocation = \frac{Nuclear_{F510}}{Whole\ cell_{F510}}$$

Further, confocal microscopy was also performed using a Carl Zeiss LSM 880 laser scanning confocal microscope at 63× (with oil) magnification at the indicated time points to demonstrate nuclear translocation of NFAT2 in B16 cells with knockdown of MCU in response to 100 μm histamine stimulation.

### Statistical analysis

All statistical analysis was performed using GraphPad Prism 8 software. All experiments were performed at least 3 times. Data are presented as mean ± SEM. Either unpaired Student's *t* test or one-sample *t* test was performed for determining statistical significance between 2 experimental samples, whereas one-way ANOVA was performed for the comparison of 3 samples. A *p*-value $<0.05$ was considered as significant and is presented as "*"; *p*-value $<0.01$ is presented as "**"; *p*-value $<0.001$ is presented as "***" and $p < 0.0001$ is presented as "****".

### Supporting information

**S1 Fig. Supporting main Fig 1. Mitochondrial $Ca^{2+}$ uptake is positively associated with melanogenesis. (A)** Representative confocal microscopy images of Mitotracker-Red staining of mitochondria on LD day 0, LD day 4, and LD day 6 B16 cells. Images have been captured at 63× (oil) magnification in a Zeiss confocal microscope (scale = 20 μm). **(B)** Quantitation of Mitotracker-Red staining to evaluate mitochondrial content on LD day 0, LD day 4, and LD day 6 B16 cells. (C) qRT-PCR analysis showing relative mRNA expression of MCU complex components (MCUb, MICU1, MICU2, and EMRE) in B16 LD model on LD day 4 and LD day 6 ($N$ = 3–5). (D) Quantitation of Rhod-2/mt-Green in LP and DP primary human melanocytes stimulated with 100 μm histamine where "*n*" denotes the number of ROIs. (E) Representative

confocal microscopy images of Mitotracker-Red staining of mitochondria in LP and DP primary human melanocytes. Images have been captured at 63× (oil) magnification in a Zeiss confocal microscope (scale = 20 μm). (F) Quantitation of Mitotracker-Red staining to evaluate mitochondrial content in LP and DP primary human melanocytes. (G) qRT-PCR analysis showing relative mRNA expression of MCU complex components (MCU, MCUb, MICU1, MICU2, and EMRE) in DP primary human melanocytes in comparison to LP primary human melanocytes ($N = 4$). Data presented are mean ± SEM. For statistical analysis, unpaired Student's $t$ test was performed for panels D and F while one-way ANOVA followed by Tukey's post hoc test was performed for panel B using GraphPad Prism software. Here, ns means non-significant; $^*$ $p < 0.05$; $^{**}$ $p < 0.01$; $^{***}$ $p < 0.001$, and $^{****}$ $p < 0.0001$. The data underlying for panels B, C, D, F, and G shown in the figure can be found in S2 Data.
(TIF)

**S2 Fig. Supporting main Fig 2. MCU positively regulates melanogenesis while MCUb negatively controls melanogenesis.** (A) qRT-PCR analysis showing decrease in MCU mRNA expression upon MCU silencing in B16 cells ($N = 4$). (B) Densitometric quantitation showing MCU levels on LD day 6 in siNT control and siMCU condition ($N = 3$). (C) Quantitation of resting mitochondrial $Ca^{2+}$ with CEPIA2mt in siNT control and siMCU B16 cells stimulated with 100 μm histamine where "$n$" denotes the number of ROIs. (D) Quantitation of resting mitochondrial $Ca^{2+}$ with CEPIA2mt in siNT control and siMCU B16 cells stimulated with 1 μm αMSH where "$n$" denotes the number of ROIs. (E) Representative traces of Fura-2 imaging to measure cytosolic $Ca^{2+}$ in siNon-Targeting (siNT) control and siMCU B16 cells stimulated with 100 μm histamine. (F) Quantitation of cytosolic $Ca^{2+}$ levels in siNT control and siMCU B16 cells stimulated with 100 μm histamine where "$n$" denotes the number of ROIs (cytosolic $Ca^{2+}$ levels in siNT control, siMCU and siMCUb were measured on same day). (G) Representative traces of Fura-2 imaging to measure cytosolic $Ca^{2+}$ in siNon-Targeting (siNT) control and siMCU B16 cells stimulated with 1 μm αMSH. (H) Quantitation of cytosolic $Ca^{2+}$ levels in siNT control and siMCU B16 cells stimulated with 1 μm αMSH where "$n$" denotes the number of ROIs. (I) Representative GFP and bright field images showing MCU-GFP transfected B16 cells (scale = 100 μm) ($N = 3$). (J) Representative western blot demonstrating ectopic expression of GFP-tagged human MCU in B16 cells ($N = 3$). (K) Representative mitochondrial $Ca^{2+}$ imaging traces of pcDNA control plasmid and MCU-GFP overexpressing B16 cells stimulated with 100 μm histamine. (L) Quantitation of mitochondrial $Ca^{2+}$ uptake by calculating increase in CEPIA2mt signal (ΔCEPIA2mt) in pcDNA control plasmid and MCU-GFP overexpressing B16 cells upon stimulation with 100 μm histamine where "$n$" denotes the number of ROIs. (M) Representative mitochondrial $Ca^{2+}$ imaging traces of pcDNA control plasmid and MCU-GFP overexpressing B16 cells stimulated with 1 μm αMSH. (N) Quantitation of mitochondrial $Ca^{2+}$ uptake by calculating increase in CEPIA2mt signal (ΔCEPIA2mt) in pcDNA control plasmid and MCU-GFP overexpressing B16 cells upon stimulation with 1 μm αMSH where "$n$" denotes the number of ROIs. Data presented are mean ± SEM. For statistical analysis, unpaired Student's $t$ test was performed for panels C, D, F, H, L, and N while one sample $t$ test was performed for panels A and B using GraphPad Prism software. $^*$ $p < 0.05$; $^{**}$ $p < 0.01$; $^{***}$ $p < 0.001$; and $^{****}$ $p < 0.0001$. The data underlying for panels A–H and K–N shown in the figure can be found in S2 Data.
(TIF)

**S3 Fig. Supporting main Fig 2. MCU positively regulates melanogenesis while MCUb negatively controls melanogenesis.** (A) Representative traces of Fura-2 imaging to measure cytosolic $Ca^{2+}$ in pcDNA control plasmid and MCU-GFP overexpressing B16 cells stimulated with 100 μm histamine. (B) Quantitation of cytosolic $Ca^{2+}$ levels in pcDNA control plasmid and

MCU-GFP overexpressing B16 cells stimulated with 100 μm histamine where "$n$" denotes the number of ROIs. **(C)** Quantitation of cytosolic $Ca^{2+}$ levels in pcDNA control plasmid and MCU-GFP overexpressing B16 cells stimulated with 1 μm αMSH where "$n$" denotes the number of ROIs. **(D)** qRT–PCR analysis showing relative mRNA expression of MCU complex components (MCUb, MICU1, MICU2, and EMRE) upon MCU-GFP overexpression. **(E)** qRT-PCR analysis showing decrease in MCU mRNA expression upon MCU silencing in primary human melanocytes ($N = 4$). **(F)** Representative western blot confirming siRNA based silencing of MCU in primary human melanocytes. Densitometric analysis using ImageJ is presented below the blot ($N = 3$). **(G)** Densitometric quantitation showing MCU levels in siNT control and siMCU condition in primary human melanocytes ($N = 3$). **(H)** Representative pellet images of siNT control and siMCU in primary human melanocytes ($N = 4$). **(I)** Mean pixel intensity of siNT control and siMCU pellet images in primary human melanocytes ($N = 3$). **(J)** qRT-PCR analysis showing decrease in Tyrosinase, TYRP1, and DCT mRNA expression upon MCU silencing in primary human melanocytes ($N = 4$). Data presented are mean ± SEM. For statistical analysis, unpaired Student's $t$ test was performed for panels B and C while one sample $t$ test was performed for panels E, G, I, and J using GraphPad Prism software. * $p < 0.05$; ** $p < 0.01$; and **** $p < 0.0001$. The data underlying for panels A–E, G, I, and J shown in the figure can be found in S2 Data.
(TIF)

**S4 Fig. Supporting main Fig 2. MCU positively regulates melanogenesis while MCUb negatively controls melanogenesis. (A)** Quantitation of resting mitochondrial $Ca^{2+}$ with CEPIA2mt in siNT control and siMCUb B16 cells stimulated with 100 μm histamine. Here, "$n$" denotes the number of ROIs. **(B)** Quantitation of resting mitochondrial $Ca^{2+}$ with CEPIA2mt in siNT control and siMCUb B16 cells stimulated with 1 μm αMSH. Here, "$n$" denotes the number of ROIs. **(C)** Representative traces of Fura-2 imaging to measure cytosolic $Ca^{2+}$ in siNon-Targeting (siNT) control and siMCUb B16 cells stimulated with 100 μm histamine. **(D)** Quantitation of cytosolic $Ca^{2+}$ levels in siNT control and siMCUb B16 cells stimulated with 100 μm histamine where "$n$" denotes the number of ROIs (cytosolic $Ca^{2+}$ levels in siNT control, siMCU and siMCUb were measured on same day). **(E)** Representative traces of Fura-2 imaging to measure cytosolic $Ca^{2+}$ in siNon-Targeting (siNT) control and siMCUb B16 cells stimulated with 1 μm αMSH. **(F)** Quantitation of cytosolic $Ca^{2+}$ levels in siNT control and siMCUb B16 cells stimulated with 1 μm αMSH where "$n$" denotes the number of ROIs. **(G)** Densitometric quantitation showing GP100 levels on LD day 6 in siNT control and siMCUb condition ($N = 3$). **(H)** Densitometric quantitation showing DCT levels on LD day 6 in siNT control and siMCUb condition ($N = 3$). **(I)** DOPA assay showing activity of tyrosinase enzyme ($N = 4$) and representative western blot for tyrosinase expression ($N = 3$) on LD day 6 upon MCUb silencing as compared to non-targeting control. Densitometric analysis using ImageJ is presented below the blot. **(J)** Densitometric quantitation showing activity of tyrosinase enzyme on LD day 6 in siNT control and siMCUb condition ($N = 4$). **(K)** Densitometric quantitation showing Tyrosinase levels on LD day 6 in siNT control and siMCUb condition ($N = 3$). **(L)** qRT-PCR analysis showing decrease in MCUb mRNA expression upon MCUb silencing in LP primary human melanocytes ($N = 3$). **(M)** Representative pellet images of siNT control and siMCUb in LP primary human melanocytes ($N = 3$). **(N)** qRT-PCR analysis showing increase in GP100, Tyrosinase and DCT mRNA expression upon MCUb silencing in primary human melanocytes ($N = 3–4$). Data presented are mean ± SEM. For statistical analysis, unpaired Student's $t$ test was performed for panels A, B, D, and F, while one sample $t$ test was performed for panels G, H, J, K, L, and N using GraphPad Prism software. Here, * $p < 0.05$; ** $p < 0.01$; *** $p < 0.001$; and **** $p < 0.0001$. The data underlying for panels A–H, J, K, L, and

N shown in the figure can be found in S2 Data.
(TIF)

**S5 Fig. Supporting main Fig 2. MCU positively regulates melanogenesis while MCUb negatively controls melanogenesis. (A)** Oxygen consumption rate (OCR) in control, MCU and MCUb overexpressing B16 cells (2 independent biologicals, with 3 technical replicates in each set). **(B)** Quantitative analysis of basal respiration in control, MCU and MCUb overexpressing B16 cells. **(C)** Quantitative analysis of ATP production in control, MCU and MCUb overexpressing B16 cells. **(D)** Quantitative analysis of maximal respiration in control, MCU and MCUb overexpressing B16 cells. **(E)** Extracellular acidification rate (ECAR) in control, MCU and MCUb overexpressing B16 cells. **(F)** Quantitative analysis of basal ECAR in control, MCU and MCUb overexpressing B16 cells. **(G)** Quantitative analysis of ECAR in control, MCU and MCUb overexpressing B16 cells post oligomycin. **(H)** Quantitative analysis of ECAR in control, MCU and MCUb overexpressing B16 cells post FCCP. **(I)** Quantitative analysis of ECAR in control, MCU and MCUb overexpressing B16 cells post rotenone/antimycin A. **(J)** Resting mitochondrial membrane potential ($\Delta\Psi$), measured with TMRE in siNT control, siMCU and siMCUb B16 cells ($N = 5$). **(K)** Mitochondrial ATP levels in siNT control, siMCU and siMCUb B16 cells ($N = 4$). Data presented are mean ± SEM. For statistical analysis, Kruskal–Wallis test was performed for panels B, C, D, F, G, I, and H, while one-way ANOVA followed by Dunnette's multiple comparisons test was performed for panel J and K using GraphPad Prism software. Here, ns means nonsignificant. The data underlying for panels A–K shown in the figure can be found in S2 Data.
(TIF)

**S6 Fig. Supporting main Fig 3. MCU regulates pigmentation in vivo. (A)** Densitometric quantitation showing MCU levels in control MO and MCU MO ($N = 3$). **(B)** The hypopigmentation phenotype analyzed at 30 hpf in around 200 zebrafish embryos from 3 independent sets of injections ($N = 3$ independent experiments with approximately 200 embryos/condition). **(C)** Representative bright-field images of zebrafish embryos injected with either control or human MCU RNA at 48 hpf ($N = 3$ independent experiments with approximately 200 embryos/condition). **(D)** Melanin-content estimation of zebrafish embryos injected with either control or human MCU RNA injection in 60 zebrafish embryos from 3 independent sets of injections ($N = 3$ independent experiments with 60 embryos/condition). Data presented are mean ± SEM. For statistical analysis, unpaired Student's $t$ test was performed for panel B, while one sample $t$ test was performed for panels A and D using GraphPad Prism software. Here, * $p < 0.05$; **** $p < 0.0001$. The data underlying for panels A, B, and D shown in the figure can be found in S2 Data.
(TIF)

**S7 Fig. Supporting main Fig 4. Transcriptomics identifies keratin filaments working downstream of mitochondrial Ca$^{2+}$ dynamics to regulate melanogenesis. (A)** Heatmap representing the expression of MCU and MCUb upon silencing of MCU and MCUb, respectively. Scale from blue to red represents z-score for fold change from −1 to +1. (B) Common oppositely regulated pathways down in siMCU and up in siMCUb. (C) qRT-PCR analysis showing decrease in keratin 5 mRNA expression upon keratin 5 silencing in B16 cells ($N = 3$). (D) qRT-PCR analysis showing decrease in keratin 7 mRNA expression upon keratin 7 silencing in B16 cells ($N = 3$). (E) qRT-PCR analysis showing decrease in keratin 8 mRNA expression upon keratin 8 silencing in B16 cells ($N = 3$). (F) qRT-PCR analysis showing decrease in Keratin 5 mRNA expression upon Keratin 5 silencing in LP primary human melanocytes ($N = 3$). (G) Representative pellet images of siNT control and siKeratin 5 in LP primary human

melanocytes ($N = 3$). Data presented are mean ± SEM. For statistical analysis, one sample $t$ test was performed for panels C–F using GraphPad Prism software. Here, ** $p < 0.01$ and *** $p < 0.001$. The data underlying for panels A–F shown in the figure can be found in S2 Data.
(TIF)

**S8 Fig. Supporting main Fig 5. Keratin 5 regulates mitochondrial Ca$^{2+}$ uptake, melanosome maturation, and positioning. (A)** Quantitation of resting mitochondrial Ca$^{2+}$ with CEPIA2mt in mCherry empty vector (EV) control and Keratin 5 overexpressing (OE) cells where "$n$" denotes the number of ROIs. **(B)** Quantitation of mitochondrial Ca$^{2+}$ uptake by calculating increase in CEPIA2mt signal (ΔCEPIA2mt) in mCherry empty vector (EV) control and Keratin 5 overexpressing (OE) cells upon stimulation with 100 μm histamine where "$n$" denotes the number of ROIs. **(C)** Quantitation of resting mitochondrial Ca$^{2+}$ with CEPIA2mt in siNT control and siKeratin 5 cells where "$n$" denotes the number of ROIs. **(D)** Quantitation of mitochondrial Ca$^{2+}$ uptake by calculating increase in CEPIA2mt signal (ΔCEPIA2mt) in siNT control and siKeratin 5 cells upon stimulation with 100 μm histamine where "$n$" denotes the number of ROIs. Data presented are mean ± SEM. For statistical analysis, unpaired Student's $t$ test was performed for panels A–D using GraphPad Prism software. Here, ** $p < 0.01$; **** $p < 0.0001$. The data underlying for panels A–D shown in the figure can be found in S2 Data.
(TIF)

**S9 Fig. Supporting main Fig 6. NFAT2 connects MCU to keratins expression. (A)** Multispecies sequence alignment of putative NFAT2 binding sites in the mouse keratin 7 (KRT7) core promoter. **(B)** Schematic representation of putative NFAT2 binding sites in the mouse KRT7 core promoter. **(C)** Multispecies sequence alignment of putative NFAT2 binding sites in the mouse keratin 8 (KRT8) core promoter. **(D)** Schematic representation of putative NFAT2 binding sites in the mouse KRT8 core promoter.
(TIF)

**S10 Fig. Time lapse images of mito Ca$^{2+}$ uptake in response to histamine and αMSH (using Cepia2mt) in B16 cells. (A)** Time lapse images of mitochondrial Ca$^{2+}$ uptake with CEPIA2mt in siNT control and siMCU B16 cells stimulated with 100 μm histamine (scale = 50 μm). **(B)** Time lapse images of mitochondrial Ca$^{2+}$ uptake with CEPIA2mt in siNT control and siMCU B16 cells stimulated with 1 μm αMSH (scale = 50 μm). Arrows indicate change in Cepia2mt signal in cells.
(TIF)

**S11 Fig. Time lapse images of mitochondrial Ca$^{2+}$ uptake in response to histamine (using Rhod-2 AM) in primary human melanocytes. (A)** Time lapse images of mitochondrial Ca$^{2+}$ uptake with Rhod-2 AM in LP and DP primary melanocytes stimulated with 100 μm histamine (scale = 50 μm). Arrows indicate change in Rhod 2AM signal in cells.
(TIF)

**S1 Data. All numerical values underlying the Figs 1–7.**
(XLSX)

**S2 Data. All numerical values underlying the S1–S8 Figs.**
(XLSX)

**S1 Raw Images. Raw images of Figs 1G, 1O, 2B, 2S, 3A, 5I, S2J, S3F and S4I.**
(PDF)

## Acknowledgments

The authors thank members of the Motiani laboratory for discussions and critical reading of the manuscript. JT acknowledges her INSPIRE Faculty Fellowship (IFA22-LSBM269) from DST, India. KA acknowledges her Junior Research Fellowship from DBT, India. The technical assistance of Ms. Nutan Sharma, Ms. Samriddhi Arora, Ms. Jaya Bharti Singh, and Mr. Shyamveer is highly appreciated.

## Author Contributions

**Conceptualization:** Rajender K. Motiani.

**Formal analysis:** Jyoti Tanwar, Kriti Ahuja, Akshay Sharma, Paras Sehgal, Gyan Ranjan, Farina Sultan, Anushka Agrawal, Donato D'Angelo, Anshu Priya, Archana Singh, Anna Raffaello, Rajender K. Motiani.

**Funding acquisition:** Rajender K. Motiani.

**Investigation:** Jyoti Tanwar, Kriti Ahuja, Akshay Sharma, Paras Sehgal, Gyan Ranjan, Anushka Agrawal, Donato D'Angelo, Anshu Priya.

**Methodology:** Jyoti Tanwar, Kriti Ahuja, Akshay Sharma.

**Project administration:** Rajender K. Motiani.

**Resources:** Vamsi K. Yenamandra, Muniswamy Madesh, Rosario Rizzuto, Sridhar Sivasubbu.

**Supervision:** Rajender K. Motiani.

**Validation:** Farina Sultan.

**Visualization:** Jyoti Tanwar, Kriti Ahuja, Akshay Sharma, Donato D'Angelo, Anshu Priya, Archana Singh, Anna Raffaello, Rajender K. Motiani.

**Writing – original draft:** Jyoti Tanwar, Rajender K. Motiani.

**Writing – review & editing:** Rajender K. Motiani.

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
