## [Editor Report · Decision Letter 0]

22 Nov 2023

Dear Dr Motiani, 

Thank you for submitting your manuscript entitled "A novel MCU-NFAT2-Keratin5 mediated feedback loop is a critical regulator of mitochondrial calcium uptake and vertebrate pigmentation" for consideration as a Research Article by PLOS Biology.

Your manuscript has now been evaluated by the PLOS Biology editorial staff as well as by an academic editor with relevant expertise and I am writing to let you know that we would like to send your submission out for external peer review.

Once your full submission is complete, your paper will undergo a series of checks in preparation for peer review. After your manuscript has passed the checks it will be sent out for review. To provide the metadata for your submission, please Login to Editorial Manager (https://www.editorialmanager.com/pbiology) within two working days, i.e. by Nov 24 2023 11:59PM.

Kind regards,

Ines

--

Ines Alvarez-Garcia, PhD

Senior Editor

PLOS Biology

---

## [Decision Letter · Decision Letter 1]

15 Feb 2024

Dear Dr Motiani,

Thank you for your patience while your manuscript entitled "A novel MCU-NFAT2-Keratin5 mediated feedback loop is a critical regulator of mitochondrial calcium uptake and vertebrate pigmentation" was peer-reviewed at PLOS Biology. Please accept my apologies for the delay in providing you with our decision. The manuscript has now been evaluated by the PLOS Biology editors, an Academic Editor with relevant expertise, and by four independent reviewers. 

As you will see, the reviewers find the conclusions interesting and novel, however they also think the data doesn’t support some of the main conclusions and recommend several experiments to address this. Both Reviewers 1 and 2 are not convinced about the model and think it has some contradictions and shortcomings. Thus, we do think it would be better to focus the manuscript on clearly demonstrating that mitochondrial Ca2+ uptake is playing a salient role in regulating melanogenesis. Reviewer 2 also thinks that the mouse data are weak and incomplete because MCU knockouts have not been analysed, and that you should do so. Reviewer 3 points out that the fly data also needs additional experiments to confirm the main conclusions and that the overall effect should be demonstrated in at least one organism. Reviewer 4, along with Reviewer 2, mentions that you should demonstrate further that the key mechanism being proposed depends on NFAT regulation. This reviewer and Reviewer 1 also think that the Ca2+-imaging experiments need concomitant readouts of cytosolic Ca2+ and that it is crucial to demonstrate the phenomenon with MSH, a bonafide melanogenic signal. Please note that histamine-induced Ca2+ signals may not be relevant to the biology being explored here and can only be considered as ancillary data to MSH-induced Ca2+ signaling. Finally, Reviewer 4 points out that mitochondrial Ca2+ imaging with Rhod2AM is prone to artifacts and that the current standard is to use a mitochondrially-targeted GECI such as CEPIA3mt. Considering that core Ca2+ signaling phenomenon needs to be demonstrated with MSH, this should be also addressed.

In light of the reviews, we would like to invite you to revise the work to thoroughly address the reviewers' reports using the guidelines mentioned above. Given the extent of revision needed, we cannot make a decision about publication until we have seen the revised manuscript and your response to the reviewers' comments. Your revised manuscript is likely to be sent for further evaluation by all or a subset of the reviewers.

**IMPORTANT - SUBMITTING YOUR REVISION**

3. Resubmission Checklist

a) *PLOS Data Policy*

b) *Published Peer Review*

Sincerely,

Ines

--

Ines Alvarez-Garcia, PhD

Senior Editor

PLOS Biology

Reviewers' comments

Rev. 1:

Mitochondrial calcium uptake is an important regulator of cellular calcium signaling, mitochondrial metabolism, and also has well-appreciated roles in regulation of cell cycle, gene expression and dell death. MCU protein mediates calcium entry into the mitochondria. Its homolog MCUb is a negative regulator of mitochondrial calcium uptake.

In this manuscript, the authors investigate the role of mitochondrial calcium dynamics in vertebrate pigmentation. The authors claim that reduced mitochondrial calcium uptake (MCU knock down) leads to reduced pigmentation, and increased mitochondrial calcium uptake (MCUb knock down) leads to increased pigmentation. They then identify Keratin 5 as a transcript that is induced by MCU knock down, and decreased by MCUb knock down, through NFAT2 transcription factor. They also claim Keratin 5 stimulates mitochondrial calcium uptake.

The role of mitochondrial calcium uptake in pigmentation is not something that has been investigated, and is of interest to the mitochondrial community. However, the data shown in this manuscript does not support some of the main conclusions and arguments of the authors (see below). As a result, re-evaluation of the mechanism by which MCU regulates pigmentation and the model proposed by the authors is needed. In addition, some important controls are missing as indicated below.

General comments:

The main weakness of this manuscript is the fact that MCU knock down increases Keratin 5 expression, yet decreases pigmentation, despite the necessity of Keratin 5 for pigmentation.

The authors try to explain this discrepancy by suggesting that Keratin 5 regulates mitochondrial calcium uptake, however, the evidence for this regulation is very weak in the manuscript. Instead, their data is consistent with a model where MCU and Keratin 5 regulate pigmentation through independent, parallel pathways, and both are required for pigmentation. Instead of trying to force MCU and Keratin 5 in one regulatory pathway (MCU --| NFAT2� Keratin 5 � MCU) the authors should acknowledge that they are not necessarily on a linear pathway as the title of the manuscript suggests. If this was true, MCU knock down should have increase pigmentation and not decrease it.

Specific comments:

(1) Please use "photo" or "image" instead of "picture" while referring to the photos of the cell pellets.

(2) Please explain how melanin content was measured and normalized, as opposed to referring to another paper. What is the melanin content normalized to? MCU knockdown can reduce cell growth (PMID: 31040260), did the authors correct for possibly lower cell number in these experiments? What was the zebrafish data normalized to? These are very important details to be able to conclude that MCU affects pigmentation, and not cell growth. What are the units on melanin graphs (ug/ 5lac cells)? What is 5lac? Please correct ug to μg where necessary.

(3) Data presented in Figs. 1C, 1D, 1G are not enough to conclude that LD day6 or DP have higher mitochondrial calcium activity. The results could be an indication of increased histamine receptors, and increased calcium release from the ER, rather than increased mitochondrial calcium uptake activity. The authors should also monitor cytosolic calcium levels under these conditions at the very least.

(4) What happens to mitochondrial content during pigmentation? Is the increased MCU activity in LD day6 and DP cells due to more mitochondria? How specific is the MCU activity increase. Please add a western blot showing MCU levels, as well as 1-2 other mitochondrial protein markers. Otherwise, it is hard to conclude that MCU activity increases, and not mitochondrial content increases overall.

(5) In Fig 3G, please show the tails against the same background. The differences in pigmentation are not easy to see by eye, and the fact the one tail is on a darker background makes it hard to compare them. Both tails should be on a white background ideally.

(6) Figs. 5B and 5C show "representative" images, but without proper statistical analysis of mitochondrial calcium levels in response to Keratin 5 overexpression, it is not possible to conclude that Keratin 5 affects mitochondrial calcium levels. How many times was this experiment done, what is the noise in the experiment? Are these changes significant? The affect sized are so small, the claim that Keratin 5 regulates MCU function is not very convincing.

(7) What are the red labelled nucleotides in Fig 6B? The authors claim that Keratin 5 promoter has consensus NFAT2 motifs (TTCC?), but their promoter alignment does not show this consensus. How good of a match are these sites, what metric is used? Please indicate any statistical evaluation/ score for NFAT2 consensus sequences in Keratin 5 promoter.

(8) The authors lack proper controls for the experiments shown in Figs. G and H. Would co-expression of any transcription factor with their lacZ reporter cause an increase? How specific is this signal. Empty vector is not good control, the authors should use another TFs and convince the reader that the increased LacZ signal they observe is specific it NFAT2, and is not observed due to overexpression of any transcription factor.

Rev. 2:

SUMMARY: The submitted manuscript explores the role of MCU and mitochondrial Ca2+ uptake in skin pigmentation. The authors claim that pigmentation requires mitochondrial Ca2+ uptake and implicate NFAT2 and Keratins as downstream components of this process. These claims are not well substantiated and there are major weaknesses in this manuscript. The mechanistic framework proposed is also weak and some aspects are incoherent. I have summarized my concerns below.

MAJOR CONCERNS:

1. MCU's role in mammalian skin pigmentation seems very modest. In B16 cells, the knockdown of MCU results in an <20% decrease in melanin production (Fig. 2F). In primary melanocytes, MCU knockdown also produced a very modest effect (n=1, no quantification, Fig S1K). Authors show increased pigmentation in the tails of gene-edited mice but the C97A mutation has not been adequately characterized through electrophysiology of mitoplasts and it's not clear if melanocytes derived from these mice indeed show substantially increased mitochondrial Ca2+ uptake. Also note that the quantification of this data is not in accord with best practice - the sample size has been made deceptively high by conflating multiple measurements from each mouse tail, with an actual total of only 3 mice (Fig. 3H). The statistical significance of such analyses cannot be taken too seriously - I think authors should use average readings from each mouse tail as a single datapoint and increase the sample size of the mice to >7 for optimal statistical power. Note that hyperpigmentation is not at all obvious in the representative image (Fig. 3G). Pointing out these weak data may seem like nitpicking, but this central claim of the manuscript has been made without the most direct experiment for such a conclusion. The crucial experiment that would establish this conclusively is the demonstration of pigmentation defects in Melanocyte-specific knockout mice of MCU. Note that MCU global knockouts are viable in some backgrounds, and no one has reported any perceptible pigmentation defects in these mice. The data presented in this manuscript did not convince me that MCU plays an especially salient role in mammalian skin pigmentation. I found the data on fly pigmentation convincing, but all further analyses are on mammalian skin pigmentation.

2. Authors have previously reported that in the B16 cell line MSH-induced pigmentation depends on Ca2+ signaling and that MSH triggers the release of ER Ca2+. But in this paper, all the measurements of mitochondrial Ca2+ are in response to Histamine, which is perplexing because to the best of my understanding, Histamine is not a melanogenic stimulus. This seems conceptually disconnected because it would be important to know if MSH stimulates the uptake of mitochondrial Ca2+ and how this might affect cytosolic Ca2+ signals. The differences seen in response to Histamine, e.g. Fig. 1C, could simply be because the cells upregulate the Histamine receptors during the 6 days of MSH-induced pigmentation protocol. Crucially, authors do not show corresponding cytosolic Ca2+ signals - this is problematic. This same issue also applies to: (1) differences in primary melanocytes of Caucasian and African origin (Fig. 1G-H); (2) analyses of MCU and MCUb knockdown and overexpression in Fig. 3.

3. Authors have proposed a mechanistic model wherein mitochondria Ca2+ uptake activates NFAT2. How does that happen? Activation of NFAT2 requires a sustained increase in cytosolic Ca2+ and downstream activation of calcineurin (which dephosphorylates NFAT). But authors have not dissected these aspects at all. First, NFAT2 translocation needs to be shown in response to MSH (not Histamine). Second, if NFAT2 activation is indeed crucial for melanogenesis, then cyclosporine, a highly potent nanomolar inhibitor of calcineurin should prevent MSH-induced pigmentation when used at 10 nM. However, it should be noted that cyclosporine is a widely used drug in transplant patients and despite long-term treatment with cyclosporine, these patients do not show any loss of pigmentation. In fact, there are case reports of hyper-pigmentation in patients taking cyclosporine. These issues again lead me to believe that the claim of MCU in mammalian skin pigmentation may have been overstated.

4. If one supposes that mitochondrial Ca2+ is a significant regulator of pigmentation, then the underlying mechanism proposed by the authors seems incoherent because there is no direct link to pigmentation machinery. Authors show some data suggesting that K5 regulates MCU, but this does not really address the key issue - how does mitochondrial Ca2+ uptake affect melanogenesis? The feedback model discussed by the authors is very confusing and does not add much to the manuscript. Perhaps there are other transcription factors that are activated by increase in cytosolic Ca2+ when mitochondria do not buffer the Ca2+ but this has not been explored.

MINOR CONCERNS

5. The implicit claim is that during melanogenesis, the mitochondrial Ca2+ uptake machinery is activated or enhanced but the use of CEPIA and Rhod-2AM also integrates upstream changes in the mobilization of cytosolic Ca2+. Quantification of mitochondrial Ca2+ uptake capacity in isolated mitochondria or digitonin permeabilized cells will get around that but authors have not carried out these studies. Why not include these direct measurements?

6. In a related aspect, please show the time lapse images of mitochondrial Ca2+ uptake in response to aMSH and Histamine (using CEPIA and Rhod2AM). These can be in the supplementary figures and the entire series need not be shown.

7. The effect of mitoxantrone on MSH-induced mitochondrial uptake and cytosolic Ca2+ signals should be shown in melanocytes. This drug was reported to inhibit MCU >5y ago but many investigators have failed to replicate these findings and currently, there is not a consensus that it inhibits MCU. Also, it would be good to check if there are credible case reports of Mitoxantrone resulting in loss of pigmentation in MS patients who take it for years. I could not find any.

Rev. 3:

In this report, Motiani and colleagues report on the function in the mitochondrial Ca2+ uniporter (MCU) in vertebrate melanogenesis.

The authors used cell lines, primary cells and a model of pigmentation in zebrafish and report that that MCU is required for melanogenesis with MCUb opposing the effects of MCU. They also briefly report that a mouse model of MCU gain of function knock-in has enhanced pigmentation. Data presented with MCU knockdown on cells in consistent with downstream activation of NFAT2 upon MCU silencing and subsequent expression of keratin 5, 7 and 8 filaments. Silencing of Keratin5 seems to augment mitochondrial Ca2+ uptake and enhance melanogenesis, suggesting that MCU-NFAT2-Keratin is part of a regulatory feedback loop that regulates melanogenesis. The authors showed that mitoxantrone, an inhibitor of MCU, inhibits pigmentation. The manuscript is overall clearly written and is relatively easy to follow but further attention to the text for redundancies would greatly enhances the clarity of the text. The results reported are certainly novel by identifying MCU as an important determinant for melanogenesis and the identification of the NFAT2/Keratin axis. The authors show that knockdown of MCU activates NFAT2, which is interesting. However, how does MCU-mediated mitochondrial calcium uptake enhance melanogenesis and melanosome maturity/function, either through metabolic or signaling pathways is not clear. There are several controls and key experiments needed to fully support the conclusions of this manuscript. I have the following specific comments for the authors to address:

1) What are the origin of B16 cells? Caucasian? and how they compare to a primary cell in term of pigmentation? This information in critical to results interpretation

2) Are the primary human melanocytes (Fig 1E)also seeded at low density? If so, what happens when you perform a similar time course assay as in Fig 1 A-B

3) Fig 2. Does MCU overexpression leads to increase in mitochondrial calcium uptake? This is important. If this is the case, then it is quite curious that excess overexperessed MCU proteins would be functional without accessory molecules such as EMRE and other proteins part of the complex? Are these proteins upregulated when MCU is overexpressed?

4) Data in Supplementary Fig S1F should be address at the same time as Fig 1. Or these data should move to Fig 1. How about other proteins of the MCU complex, are they also highly expressed in DP vs LP?

5) Re: DP vs LP: I think it would be more judicious to silence MCU in the DP melanocytes to see if this will affect pigmentation?

6) Along the same lines, Are the levels of MCUb different between LP and DP melanocytes? This is also important!

7) The authors should also silence and/or overexpress MCUb in DP melanocytes

8) Did the authors inject MCUb RNA in Zebrafish? These in principle should reduce melanogenesis. What about MCUb morpholinos?

9) Mice KI MCU: Show additional tails to enable visual comparisons by the reader. Show data that indeed KI cells have increased Mitochondrial Ca2+ uptake or high mitochondrial calcium at the basal level.

10) Fig 5 I: these blots are not very convincing. Please show better quality blots. Please make sure background are visible in all blots. Please provide full uncut and unprocessed blots related to all data.

11) Fig 6: Authors should mutate this NFAT2 putative promoter region and determine if luciferase activity is reduced.

Rev. 4:

The study delves into the role of MCU in pigmentation, employing both in vitro and in vivo methodologies. The authors manipulated the expression of MCU, MCUb, and keratin 5, observing the correlation between high mCa2+ levels and increased melanin synthesis. Notably, this effect was observed in a model featuring NFAT2-controlled keratin expression. The primary conclusion is that MCU regulates skin pigmentation through NFAT2, influencing keratin 5.

In summary this is a good paper addressing an important biological question.

Comments:

Major:

- The study appears oversimplified, lacking crucial controls. The authors overlook the role of SOCE as well as other mitochondrial parameters influenced by MCU, potentially involved in pigmentation control.

- Previous research on MCU, SOCE, and NFAT interplay is mainly disregarded. Given SOCE's known role in melanin synthesis and MCU's control over SOCE, further investigation into the MCU-SOCE-NFAT interplay is needed.

- The impact of genetic manipulations of MCU, MCUb, and keratin 5 on mitochondrial function and bioenergetics is not addressed.

- The role of SOCE, NFAT and MCU have been investigated in melanoma cells previously. Was there any effect on cell pigmentation reported? Does treatment of melanoma cells but also patients with calcineurin (and hence NFAT) inhibitors (tacrolimus etc.) affect their pigmentation?

Minor:

- Using murine melanoma cells for pigmentation studies is not the most appropriate approach.

- Given that alphaMSH causes calcium uptake (published by the same authors) the use of artificial stimulus such as histamine needs to be explained.

- Rhod2AM is prone to artifacts. The authors should use CEPIA or similar for all measurements of mCa2+.

- How many melanocyte lines were used for the data shown in Fig. 1 E-H? Only 1+1?

- Is the MCU expression increased in the samples shown in Fig 1A?

---

## [Decision Letter · Decision Letter 2]

2 Aug 2024

Dear Dr Motiani,

Thank you for your patience while we considered your revised manuscript entitled "Mitochondrial calcium uptake orchestrates vertebrate pigmentation via transcriptional regulation of keratin filaments" for consideration as a Research Article at PLOS Biology. Your revised study has now been evaluated by the PLOS Biology editors, the Academic Editor and the four original reviewers. 

The reviews are attached below. In light of the reviews, we are pleased to offer you the opportunity to address the remaining points from Reviewers 2 and 4 in a revision that we anticipate should not take you very long. We will then assess your revised manuscript and your response to the reviewers' comments with our Academic Editor aiming to avoid further rounds of peer-review, although might need to consult with the reviewers, depending on the nature of the revisions.

As mentioned, we would like you to perform the experiments suggested by Reviewer 4 as following:

1. Seahorse analysis to assess mitochondrial respiration and glycolytic activity.

2. ATP production assays using commercially available luminescence-based ATP assay kits 3. Mitochondrial membrane potential assessments using TMRE or JC-1 dyes (flow cytometry or microscopy).

**IMPORTANT - SUBMITTING YOUR REVISION**

Sincerely,

Ines

--

Ines Alvarez-Garcia, PhD

Senior Editor

PLOS Biology

Reviewers' comments

Rev. 1:

The authors significantly improved the manuscript during revisions. All of my initial critiques are being addressed. The new genetic data is very compelling and shows an important role for mitochondrial calcium signaling in regulation of cellular calcium dynamics and ensuing changes in gene expression and pigmentation.

Rev. 2:

Authors have addressed many of my concerns and the manuscript is significantly improved. However, the finding that 10 micromolar Cyclosporine is required to inhibit aMSH-induced pigmentation is clear evidence that Caclineurin-NFAT axis is not involved in this process. This conclusion is grossly overstated and the evidence is very weak. Note that Cyclosporine has an IC50 of 5 nM for Calcineurin. In T cells, 10-20 nM cyclosporine completely prevents NFAT translocation. Authors show clearly that even 100 nM cyclosporine has no effect on melanogenesis. They get an effect with 10 uM (1000-fold higher than IC50) where non-specific effects of cyclosporine, including on mitochondrial mechanisms are in play. Furthermore, the technical quality of NFAT translocation data does not inspire confidence. In control cells, Histamine resulted in practically no translocation despite Ca2+ elevations. These data are derived with transient transfection of NFAT-GFP and the images seem cherry-picked. In light of these concerns, my view is that the conclusion of NFAT involvement is not definitive and needs to be rolled back. Authors may suggest that possibility along with the caveats noted above. This change is especially important in the abstract. Beyond this significant reservation, I do not have a problem with this manuscript moving towards publication.

Rev. 3:

The authors have put forth a responsive revision and have addressed all my comments. No further comments.

Rev. 4:

The authors have revised their manuscript and successfully addressed most of the reviewers' comments. The overall quality of the paper has significantly improved.

One issue that still remains is a more accurate characterization of essential mitochondrial functions. While I am not insisting, I would like to encourage the authors to perform such experiments. After all, MCU is a mitochondrial protein complex, and characterizing its role in mitochondrial function will be beneficial for researchers working in the field.

Furthermore, I would suggest explaining in more detail that some of the observed effects might be due to the mitochondrial modulation of SOCE and thereby NFAT activity.

---

## [Editor Report · Decision Letter 3]

13 Sep 2024

Dear Dr Motiani,

Thank you for your patience while we considered your revised manuscript entitled "Mitochondrial calcium uptake orchestrates vertebrate pigmentation via transcriptional regulation of keratin filaments" for publication as a Research Article at PLOS Biology. This revised version of your manuscript has been evaluated by the PLOS Biology editors and the Academic Editor.

Based on our Academic Editor's assessment of your revision, we are likely to accept this manuscript for publication, provided you satisfactorily address the data and other policy-related requests stated below.

We expect to receive your revised manuscript within two weeks. 

*Published Peer Review History*

*Press*

Sincerely,

Ines

--

Ines Alvarez-Garcia, PhD

Senior Editor

PLOS Biology

DATA POLICY:

Many thanks for providing the data underlying the graphs shown in the figures. I have checked them and find some of the data missing. Please provide the data underlying the graphs in the following figures:

Fig. 1C, E, K, M; Fig. 2C, D, M, N; Fig. 3G; Fig. 4B, C; Fig. 5B, C; Fig. 7A, C; Fig. S2E, G, K, M; Fig. S3A; Fig. S4C, E; Fig. S5A, E and Fig. S7A, B

Please also ensure that figure legends in your manuscript include information on where the underlying data can be found. For example, you could add at the end of the corresponding figure legends the following: "The data underlying the graphs shown in the figure can be found in S1 Data."

**Please also make publicly available at this time the data deposited in the NCBI Sequence Read Archive (SRA) under the accession number PRJNA1112319.

CODE POLICY

Per journal policy, if you have generated any custom code during the course of this investigation, please make it available without restrictions. Please ensure that the code is sufficiently well documented and reusable, and that your Data Statement in the Editorial Manager submission system accurately describes where your code can be found. [IF APPLICABLE: As the code that you have generated to XXX is important to support the conclusions of your manuscript, its deposition is required for acceptance.]

---

## [Editor Report · Decision Letter 4]

11 Oct 2024

Dear Dr Motiani,

Thank you for the submission of your revised Research Article entitled "Mitochondrial calcium uptake orchestrates vertebrate pigmentation via transcriptional regulation of keratin filaments" for publication in PLOS Biology. On behalf of my colleagues and the Academic Editor, Bimal Desai, I am delighted to let you know that we can in principle accept your manuscript for publication, provided you address any remaining formatting and reporting issues. These will be detailed in an email you should receive within 2-3 business days from our colleagues in the journal operations team; no action is required from you until then. Please note that we will not be able to formally accept your manuscript and schedule it for publication until you have completed any requested changes.

PRESS

Sincerely, 

Ines

--

Ines Alvarez-Garcia, PhD

Senior Editor

PLOS Biology
